# Wind-driven upwelling and surface nutrient delivery in a semi-enclosed coastal sea

Ben Moore-Maley[1] and Susan E. Allen[1]

[1]Department of Earth, Ocean and Atmospheric Sciences, University of British Columbia, 2207 Main Mall, Vancouver, BC V6T 1Z4, Canada

**Correspondence:** Ben Moore-Maley (bmoorema@eoas.ubc.ca)

**Abstract.**

Wind-driven upwelling is an important control on surface nutrients and water properties in stratified lakes and seas. In this study, a high resolution biophysical coupled model is used to investigate upwelling in the Strait of Georgia. The model is forced with surface winds from a high resolution atmospheric forecast and has been tuned in previous studies to reproduce extensive observations of water level, temperature, salinity, nutrients and chlorophyll with competitive skill relative to similar models of the study region. Five years of hourly surface nitrate and temperature are analyzed in order to characterize the dominant upwelling patterns of the basin. A prevailing along-axis wind pattern steered by mountainous topography produces episodic upwelling along the western shore during the spring and fall southeasterlies and along the eastern shore during the summer northwesterlies, as indicated by positive nitrate anomalies. Principal component analysis reveals that these cross-axis upwelling patterns account for nearly one-third of the surface nitrate variance during the productive season. By contrast, nearly half of the surface temperature variance over the same period is dominated by a single, combined mixing and diurnal heating-cooling pattern. The principal components associated with these patterns correlate with along-axis wind stress in a manner consistent with these physical interpretations. The cross-axis upwelling response to wind is similar to other dynamically wide basins where the baroclinic Rossby deformation radius is smaller than the basin width. However, the nitrate anomaly during upwelling along the eastern shore is stronger in the northern basin, which may be indicative of an along-axis pycnocline tilt or an effect of the background along-axis stratification gradient due to the Fraser River. Our findings highlight an important spatio-temporal consideration for future ecosystem monitoring.

## 1 Introduction

Wind-driven upwelling is ubiquitous in the coastal ocean (Kämpf and Chapman, 2016) and in large, enclosed seas (e.g., Silvestrova et al., 2019; Lehmann et al., 2012; Plattner et al., 2006). This process provides an important pathway for vertical nutrient transport into the euphotic zone (Messié and Chavez, 2015) but can also be a source of environmental stress by introducing large amplitude fluctuations of temperature, dissolved oxygen, $pCO_2$, pH and aragonite saturation state near the coast (Chan et al., 2017). Consequently, upwelling variability can dramatically alter seasonal biological community structure across a wide trophic range (Barth et al., 2007), and persistent upwelling is often a defining feature of ecosystems such as along

Pacific and Atlantic eastern boundaries (Chavez and Messié, 2009). Additionally, upwelling has been observed to influence phytoplankton distributions in large lakes (Mziray et al., 2018; Bouffard et al., 2018) and can mitigate or exacerbate problems associated with anthropogenic eutrophication in urbanized regions such as cyanobacteria blooms in the Baltic Sea (Wasmund et al., 2012) and hypoxia in Lake Erie (Rowe et al., 2019).

In enclosed basins, coastal upwelling describes the pycnocline displacements that result from wind-driven horizontal diver-
30 gence near a coastline and any oscillations that follow (Csanady, 1977; Shintani et al., 2010). However, the structure of these pycnocline displacements strongly depends on the dynamic width of the basin (Cushman-Roisin et al., 1994). This dynamic width can be described by the dimensional width of the basin relative to the internal Rossby deformation radius $L_R$

$$L_R = \frac{NH}{f} \tag{1}$$

where $N$ is the Brunt-Väisälä frequency, $H$ is the water depth and $f$ is the Coriolis parameter. Given a sustained, along-axis
wind impulse, if the basin is much wider than $L_R$ then Ekman fluxes freely develop and the pycnocline displacement is cross-axis. If the basin is narrower than $L_R$ then Ekman fluxes are rapidly opposed by cross-axis pressure gradients and cannot fully form, forcing the surface layer transport to flow along-axis which sets up an end-to-end pycnocline tilt. Upwelling studies across a range of dynamic widths support this theory. Cross-axis upwelling is primarily observed in large seas like the Baltic Sea sub-basins (Bednorz et al., 2019; Delpeche-Ellmann et al., 2017; Zhurbas et al., 2008) and the North American Great
Lakes (Plattner et al., 2006; Csanady, 1977) while along-axis upwelling is primarily observed in narrow basins like the glacial reservoirs of British Columbia (Imam et al., 2013; Laval et al., 2008). The along-axis pycnocline setup in lakes progresses as a set of internal seiche modes excited by the initial wind impulse (Stevens and Lawrence, 1997; Stevens and Imberger, 1996). If the lake is wide enough, these seiches can be amphidromic to the point of resembling Kelvin waves, even if the initial pycnocline setup is along-axis (Roberts et al., 2021; Bouffard and Lemmin, 2013; Valerio et al., 2012).

In this study, we explore wind-driven upwelling in the Salish Sea estuarine fjord network on the Canadian west coast. We focus our analysis on the Strait of Georgia (SoG), which is the largest and most protected basin in the Salish Sea. The SoG is a globally significant ecosystem that provides critical habitat to several keystone fisheries including the iconic Fraser River sockeye salmon (Thomson et al., 2012) and the endangered Southern Resident Killer Whale (Wasser et al., 2017). The SoG also contains over half of the shellfish aquaculture leases in Pacific Canada, most of which are concentrated in just two sheltered
regions. Salinity gradients in the SoG are large due to significant runoff from surrounding mountainous watersheds and deep estuarine inflows from the Pacific Ocean. These oceanic inflows combined with local remineralization of organic matter make the deep SoG nutrient and carbon rich (Sutton et al., 2013; Ianson et al., 2016) and transiently hypoxic (Johannessen et al., 2014), while the surface is nutrient and carbon poor due to strong seasonal primary productivity. Several recent studies have sought to link changes in this productivity to the significant declines in returning salmon in recent years. While no long term
changes have been identified (Johannessen et al., 2021), interannual variability in spring bloom timing has been implicated as a possible factor (Thomson et al., 2012).

The importance of wind in controlling productivity is well-established. During the transition from winter to spring, the expansion of the North Pacific High suppresses winter storm activity in the region (Bakri and Jackson, 2019). Calm winds and

clear skies provide light and stratification for a spring phytoplankton bloom to occur (Collins et al., 2009). Surface nutrients
are rapidly depleted and remain low throughout the summer, with small phytoplankton persisting through the nutrient depleted
conditions (Del Bel Belluz et al., 2021; Haigh and Taylor, 1991). Further wind activity throughout this period introduces addi-
tional nutrients to the surface allowing smaller blooms to occur (Del Bel Belluz et al., 2021). Additionally, large surface $pCO_2$
fluctuations have been observed to occur near sensitive shellfish aquaculture zones in the northern SoG following northerly
wind impulses (Evans et al., 2018). Poor spatial and temporal resolution in most existing studies has prevented a more detailed
analysis of surface nutrient delivery in the SoG, and so these nutrient and $pCO_2$ pulses are generally attributed to vertical wind
mixing (e.g., Del Bel Belluz et al., 2021; Yin et al., 1997). However, cross-axis asymmetry of satellite sea surface temperature
(Evans et al., 2018) and modelled surface nitrate (Olson et al., 2020) identified during more recent along-axis wind events in
the northern SoG suggests that upwelling is a significant driver of these nutrient pulses.

Here we analyze 5 years of hourly, high resolution surface nitrate and temperature results from a biophysical coupled model
of the Salish Sea along with hourly, high resolution surface wind forcing fields from an operational Canadian weather model
in order to better characterize the mechanism of wind-driven surface nutrient delivery in the SoG. Using principal component
analysis, we identify dominant leading modes of variability that account for significant fractions of the total variance of each
tracer following the spring bloom. By examining the spatial patterns and power spectra of the surface nitrate and temperature
modes, and the spectral coherence and correlation between these tracer modes and the wind forcing record, we attribute the
variance from each mode to a set of physical processes including upwelling. This study is the first to quantify the contribution of
wind-driven upwelling to the variability of surface nitrate in the SoG, and the results highlight a fundamental nutrient delivery
mechanism that is currently unexplored in SoG ecosystem studies. This study also provides a further basis for interpreting
surface temperature and $pCO_2$ fluctuations like those observed in the northern SoG (Evans et al., 2018). Finally, this study
corroborates the well-established cross-axis patterns of upwelling in dynamically wide basins such as the Baltic Sea and North
American Great Lakes but for a strongly stratified estuarine fjord system.

## 2 Methods

### 2.1 Study area

The SoG is approximately 200 km long, 40 km wide and 400 m deep along the thalweg (Fig. 1). The primary connection to the
Pacific Ocean is located at the southern end of the basin through Haro and Rosario Straits while a secondary connection to the
north through Johnstone Strait accounts for a small fraction of the oceanic exchange (Pawlowicz et al., 2007). The exchange
between the SoG and the Pacific Ocean is forced by estuarine circulation (MacCready et al., 2021) fed by local watersheds
as well as the approximately 220,000 km$^2$ Fraser River drainage basin. This circulation cell is significantly modulated by
mixing due to 2-4 m amplitude mixed diurnal-semidiurnal tides (Foreman et al., 1995), primarily over the respective 100 and
50 m sill depths in the narrow Haro and Johnstone Straits. This tidal modulation recycles outgoing brackish water back into
the intermediate and deep SoG – the incoming water through Haro Strait, for example, is composed of approximately 60%

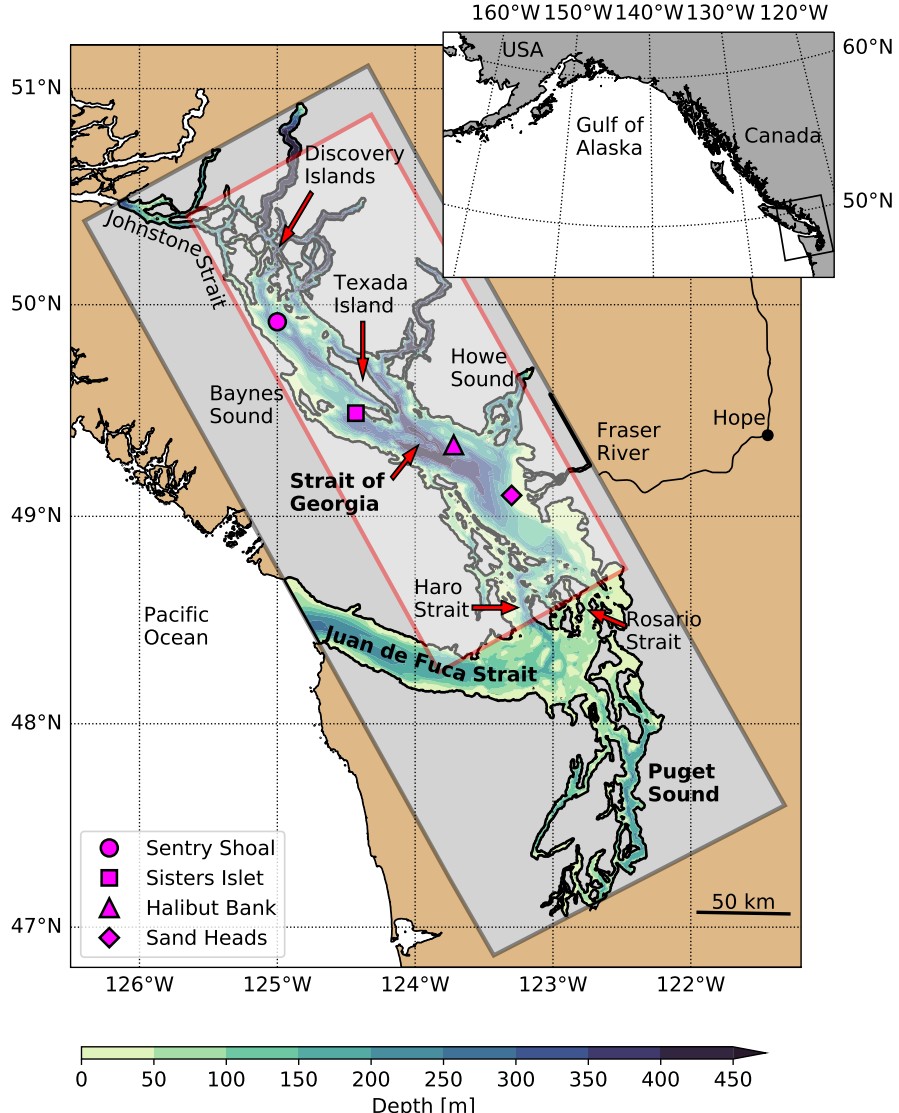

**Figure 1.** Map of the Salish Sea study region with the SalishSeaCast model domain overlaid as a gray box and contoured bathymetry. Relevant geographic features, meteorological observing platforms (magenta symbols) and the subdomain used for principal component analysis (red box) referenced in this study are also shown. Coastline data were obtained from the GSHHS database (Wessel and Smith, 1996).

outgoing surface water (Pawlowicz et al., 2019). However, annual flushing maintains high salinity in the deep basin (Masson, 2002) and strong vertical salinity gradients below the brackish surface layer.

Oceanic inflows driven by estuarine circulation are the primary sources of nitrate (Mackas and Harrison, 1997) and dissolved inorganic carbon (Ianson et al., 2016) to the SoG. Local remineralization combined with long residence times amplify these

tracers in the deep SoG and reduce deep dissolved oxygen to increasingly hypoxic levels (Johannessen et al., 2014; Ianson

et al., 2016). Seasonal cycles of wind, solar and freshwater forcing determine the nitrate and $pCO_2$ levels in the surface. During the spring expansion of the North Pacific High, storm suppression facilitates the spring phytoplankton bloom (Collins et al., 2009) which depletes surface nitrate and increases surface $pCO_2$ (Moore-Maley et al., 2016). These surface conditions persist throughout the summer, enhanced by the combined stratifying effects of solar irradiance and river runoff (Masson and Peña, 2009). The latter reaches a maximum in early summer during the Fraser River freshet and has the strongest influence in the southern SoG near the Fraser River delta (Pawlowicz et al., 2017). Strong summer wind events have been shown to briefly replenish surface nutrients (Del Bel Belluz et al., 2021) and decrease surface $pCO_2$ (Evans et al., 2018) against this background stratification. Surface nitrate is not fully depleted at the ends of the SoG near Haro and Johnstone Straits due to tidal mixing (Masson and Peña, 2009; Olson et al., 2020). The strength of this mixing varies over a fortnightly tropical tidal cycle (Thomson et al., 2020).

Wind over the SoG is determined by the synoptic scale meteorology of the northeast Pacific Ocean. Consistent groupings of three dominant synoptic types have resulted from cluster analyses of meteorological data at marine (Bakri and Jackson, 2019) and land-based (Stahl et al., 2006) observing platforms in the region. These types can be summarized as (1) a summer type with a pronounced North Pacific High and northwesterly, along-isobar wind at the coast, (2) a winter type with a pronounced Aleutian Low and strong southeasterly, cross-isobar wind at the coast and (3) a transition type between the two pressure centers that dominates during the shoulder seasons (Bakri and Jackson, 2019; Stahl et al., 2006). The strongest observed winds at marine locations are associated almost exclusively with the winter synoptic types, and spatial variability between locations is consistent with the presence of extra-tropical cyclones (Bakri and Jackson, 2019). Conversely, summer synoptic types are weaker but more persistent than winter types in terms of their recurrence probability, and they are also generally longer lived with a maximum duration of 15-20 days relative to the shorter 5-10 day maximum of the winter types (Stahl et al., 2006). The dominant northwesterly and southeasterly directions align closely with the primary axis of the SoG, and further topographic steering constrains the local wind to blow primarily along-axis (Bakri et al., 2017a). A significant exception to this climatology are the winter gap winds that occur in the Fraser River Valley and Howe Sound (Bakri et al., 2017b; Jackson, 1996; Overland and Walter Jr., 1981).

## 2.2 Numerical simulations

The SalishSeaCast model employed in this study is a quasi-operational, biophysical-coupled, Salish Sea configuration of the Nucleus for European Modelling of the Ocean (NEMO) 3.6 engine, developed and maintained by the UBC Mesoscale Ocean and Atmospheric Dynamics (MOAD) Laboratory (Olson et al., 2020; Soontiens and Allen, 2017; Soontiens et al., 2016). NEMO is a finite difference, curvilinear, hydrostatic, primitive equation ocean model with extensive functionality for coupling additional system components such as biogeochemistry, passive tracers, Lagrangian particles and sea ice (Madec et al., 2017). The SalishSeaCast domain has approximately 500 m horizontal resolution and 40 $z$-coordinate layers approximately 1 m thick near the surface, coarsening to 27 m in the bottom layer. The model uses split-explicit time stepping with barotropic and baroclinic time steps of 2 s and 40 s, respecively. A 2 s vertical time step is also specified to maintain stability in the tidal mixing regions where vertical velocities are large. Lateral mixing is constrained by constant horizontal eddy viscosity and diffusivity

values of 1.5 m$^2$ s$^{-1}$, and vertical mixing is parameterized using a $k$-$\epsilon$ turbulence closure scheme with background vertical eddy viscosity and diffusivity values of $1 \times 10^{-6}$ m$^2$ s$^{-1}$. Partial slip conditions are imposed at lateral closed boundaries, and quadratic bottom friction and a log layer parameterization are imposed at the bottom boundary.

The SalishSeaCast ecosystem model, SMELT (Salish Sea Model Ecosystem-Lower Trophic), is a nutrient-phytoplankton-zooplankton-detritus (NPZD) scheme coupled to the physics engine using the NEMO Tracers in the Ocean Paradigm (TOP) module (Olson et al., 2020). SMELT simulates the dominant functional groups of the Salish Sea, specifically: 3 nutrient classes (nitrate, ammonium, silica), 3 phytoplankton classes (diatoms, small flagellates, *Mesodinium rubrum*), 2 grazer classes (microzooplankton, mesozooplankton) and 3 detrital pools (biogenic silica, particulate organic nitrogen, dissolved organic nitrogen). *M. rubrum* is a mixotrophic ciliate that can contribute significantly to observed phytoplankton biomass in the Salish Sea (Hansen et al., 2013), and both the autotrophic and heterotrophic behaviors attributed to this ciliate are simulated in the model. While an early diatom-only version of SMELT was used to predict the spring bloom timing in the SoG (Collins et al., 2009), the multiple NPZD classes represented in the present version are necessary to accurately resolve the post-bloom dynamics that govern surface nitrate concentration (Olson et al., 2020).

Open boundaries in Juan de Fuca Strait and Johnstone Strait are forced with eight tidal constituents (K1, O1, P1, Q1, M2, K2, N2, S2) determined according to Foreman et al. (2000) and Thomson and Huggett (1980), respectively. Surface height is additionally forced at the Juan de Fuca boundary using a storm surge forecast at Neah Bay, USA provided by the US National Oceanic and Atmospheric Administration (NOAA). The Flather radiation condition (Flather, 1994) is used for barotropic tidal velocities and sea surface height, a modified Orlanski radiation condition (Marchesiello et al., 2001) with a sponge layer is used for baroclinic velocities, and the Flow Relaxation Scheme (Madec et al., 2017) is used for temperature and salinity within 10 grid points from the boundary. Temperature, salinity and nitrate at the Juan de Fuca open boundary are obtained from detided outputs of the University of Washington LiveOcean model (Fatland et al., 2016). Dissolved silica is further estimated from LiveOcean nitrate (Olson et al., 2020). Temperature, salinity, nitrate and silica at the Johnstone Strait boundary are set based on monthly climatologies provided by the Hakai Institute. Ammonium at both boundaries is set to a mean observed profile (Olson et al., 2020).

Surface momentum, heat, precipitation, and sea level pressure are forced with results from the High Resolution Deterministic Prediction System (HRDPS), which is a 2.5 km operational Pan-Canadian weather forecast model developed and maintained by Environment and Climate Change Canada (ECCC) (Milbrandt et al., 2016). Wind stress $\overrightarrow{\tau}$ is calculated according to a quadratic wind speed parameterization

$$\overrightarrow{\tau} = \rho_\mathrm{a} C_\mathrm{s} U \mathbf{U} \tag{2}$$

where $\mathbf{U}$ is the surface wind velocity vector, $U$ is the surface wind speed, $\rho_\mathrm{a}$ is the surface air density and $C_\mathrm{s}$ is the surface drag coefficient calculated according to Hellerman and Rosenstein (1983). The wind stress boundary condition is applied to the surface momentum via the vertical momentum flux equation

$$\frac{\overrightarrow{\tau}}{\rho_0} = A_v \frac{\partial \mathbf{u}_h}{\partial z}\bigg|_{z=1} \tag{3}$$

where $\mathbf{u}_h$ is the horizontal fluid velocity vector, $A_v$ is the vertical eddy diffusivity determined by the $k$-$\epsilon$ parameterization, $\rho_0$ is the background density and $z$ is the vertical coordinate.

Monthly climatologies are used to prescribe runoff from the approximately 150 rivers included in the model domain (Morrison et al., 2012). Upper Fraser River runoff is prescribed separately using daily observations provided by ECCC from a flow gauge approximately 150 km upstream at Hope, British Columbia while the lower Fraser tributaries are included in the climatologies. Temperature in all rivers is set according to a monthly climatology of Fraser River temperature based on ECCC observations at Hope (Allen and Wolfe, 2013), and river salinity is set to zero. Biological river tracers are determined according to ECCC and UBC observations in selected watersheds including the lower Fraser (Olson et al., 2020). For example, Fraser River nitrate is set to a seasonal cycle that varies between 8.37 $\mu$M N during the winter and 2.88 $\mu$M N during the freshet.

    The tidal constituents at the open boundaries have been tuned to improve model skill at 31 tidal gauges located throughout the model domain (Soontiens et al., 2016). This tuning exercise prioritized high skill in the SoG region, and the tidal predictions outside of the SoG are thus less accurate. However, tidal mixing between Juan de Fuca Strait and the SoG is sufficiently accurate to resolve deep basin flushing, which is important for controlling temperature and salinity in the SoG over time (Soontiens and Allen, 2017). Soontiens and Allen (2017) further determined that the largest improvements to deep flushing were achieved by removing a numerical kinetic energy sink artifact known as the Hollingsworth instability (Hollingsworth et al., 1983). Temperature, salinity and the biological tracers have been rigorously evaluated against observations from several comprehensive, multi-year sampling programs covering the full model domain (Olson et al., 2020). Temperature is the most accurately predicted tracer with no significant bias. Salinity is predicted nearly as accurately as temperature except at low salinity, especially near the coast where the limitations of the monthly watershed climatology become important and coastal processes like wetting and drying are unresolved in the model. Model nitrate, dissolved silica and chlorophyll all demonstrate average to strong skill relative to similar models of the study region (e.g., Khangaonkar et al., 2018), with nitrate performing the strongest of the three tracers. Surface nitrate in the SoG is biased low by approximately 1-2 $\mu$M during the productive season since model phytoplankton are slightly more effective at depleting surface nutrients than observed communities. This problem has also been encountered in previous biological modelling studies of the SoG (e. g., Moore-Maley et al., 2016).

    A 5-year hourly hindcast record of the SalishSeaCast model was obtained by performing parallel CPU runs on a high performance computing platform. A restart file was produced by initializing the model tracers according to several data sets (Olson et al., 2020) and spinning up from rest over two consecutive years of 2013 forcing data. This restart file was then used to initialize the hindcast at the beginning of 2013 and run through the end of 2019. We limit our analysis to years 2015-2019 during the period between spring and fall when surface nitrate is depleted. We define these seasonal bounds using a nitrate threshold of 2 $\mu$M applied to the spatial median of the surface nitrate concentration over an open water region of the SoG to be described in Section 3. Specifically, the analysis window begins 5 days after the surface nitrate first drops below the threshold value and ends 5 days before the surface nitrate last rises above the threshold value. The 5-day buffer is included to further eliminate the biological influence of large blooms on the upwelling signal. We refer to this period as the "productive season".

### 2.3 Data methods

**Principal component analysis**

We perform principal component analysis (PCA) as described by Preisendorfer (1988) on the hourly surface nitrate and temperature records during the productive season, and we exclude Juan de Fuca Strait and Puget Sound to remove the variance contributed by processes outside of the SoG (Fig. 1, red box). To prepare the fields for PCA, we resample the model grid to a resolution of 2.5 km in order to improve computation time and then "de-trend" the resampled nitrate and temperature records by subtracting a low-pass filtered signal with an approximate 35 d cutoff frequency. The 35 d cutoff was chosen to remove any seasonal variability still present during the productive season while retaining the processes occurring in the subtidal frequency range such as the fortnightly tidal cycle and the synoptic scale wind events introduced in Section 2.1. A finite impulse response (FIR) filter with a 1235 h Blackman window is used, which attenuates the signal 10-fold at an effective 841 h, or 35 d, cutoff frequency.

To perform the PCA matrix operations, the spatial dimensions of each tracer record are treated as a set of $p$ independent variables over a time series of length $n$ such that the filtered nitrate and temperature arrays are each reshaped to an $n \times p$ matrix $\mathbf{Z}$. The principal component (PC) loadings and empirical orthogonal functions (EOF) are obtained using the singular value decomposition of $\mathbf{Z}$

$$\mathbf{Z} = \mathbf{A}'\mathbf{L}^{1/2}\mathbf{E}^{\mathrm{T}} = \mathbf{A}\mathbf{E}^{\mathrm{T}} \tag{4}$$

where $\mathbf{A}$ is the $n \times p$ PC loadings matrix, $\mathbf{L}$ is the $p \times p$ triangular eigenvalue matrix and $\mathbf{E}$ is the $p \times p$ EOF pattern matrix (Preisendorfer, 1988).

The orthogonality constraint of PCA can limit the extent to which each mode of variability actually captures the full pattern of a given physical phenomenon (Hannachi et al., 2007). We thus introduce a $p \times p$ rotation matrix $\mathbf{R}$ such that

$$\mathbf{Z} = \mathbf{A}(\mathbf{R}\mathbf{R}^{\mathrm{T}})\mathbf{E}^{\mathrm{T}} = (\mathbf{A}\mathbf{R})(\mathbf{E}\mathbf{R})^{\mathrm{T}} = \mathbf{B}\mathbf{U}^{\mathrm{T}} \tag{5}$$

where $\mathbf{B}$ is the rotated $n \times p$ PC loading matrix and $\mathbf{U}$ is the rotated $p \times p$ EOF pattern matrix. $\mathbf{R}$ is chosen such that the varimax criterion

$$f(\mathbf{R}) = \sum_{j=1}^{p} \left\{ \frac{1}{n} \sum_{t=1}^{n} \left[ b_j^2(t) \right]^2 - \left[ \frac{1}{n} \sum_{t=1}^{n} b_j^2(t) \right]^2 \right\} \tag{6}$$

is maximized, where $b_j(t)$ are the elements of $\mathbf{B}$ (Preisendorfer, 1988). We use the Simultaneous Factor Varimax Solution algorithm described by Horst (1965) in order to find $\mathbf{R}$.

**Spectral analysis and correlation**

We define a reference, along-axis HRDPS wind stress record by rotating the meridional wind velocity 55.6° CCW, computing the wind stress according to Equation 2 and taking the spatial median over an open water region of the SoG to be described

in Section 3. We determine the power spectral density (PSD) for the PC loadings time series of a given mode and the spectral coherence between each mode and our reference, along-axis wind stress using the multitaper ensemble method (Percival and Walden, 1993). For PSD, we use a normalized half-bandwidth of 2.5 or an approximate dimensional smoothing bandwidth of $0.007 \, \mathrm{d}^{-1}$, which is sufficiently narrow to resolve any fortnightly and monthly subtidal peaks that may be present. For spectral coherence, we increase this half-bandwidth to 15 or an approximate dimensional bandwidth of $0.043 \, \mathrm{d}^{-1}$ since we are more

interested in broadly-distributed subtidal wind forcing. The maximum effective ensemble size is twice the half-bandwidth, or 5 tapers for the PSD calculation and 30 tapers for the coherence calculation. We expect wind-driven processes to demonstrate significant auto-regressive "red noise" distributions in their PSD due to the persistence observed in atmospheric phenomena (Trenberth, 1984). We therefore introduce a significance threshold to identify PSD features that deviate from this stochastic atmospheric variability such as the fortnightly tidal cycle. This threshold is defined here as the 99th percentile over 1000

randomly-seeded, first-order, auto-regressive (AR1) "red noise" processes. We further define the significance threshold for coherence as the 99th percentile of the coherence between 1000 pairs of randomly-seeded, Gaussian "white noise" signals.

    We use a Bayesian Markov Chain Monte Carlo (MCMC) method to determine the linear regression and correlation between each PC mode and the along-axis HRDPS wind stress (see Section 2.4 below). The method uses a No-U-Turn Sampler (NUTS) with a sample size of $n = 1000$. The NUTS sampler returns normal distributions of the linear regression slope and intercept.

Confidence intervals are then determined by drawing 1000 samples from these distributions, and we consider the correlation significant if the 99% confidence intervals do not include a zero slope. We define a Bayesian correlation coefficient in terms of the variances of the fit parameters

$$R^2 = \frac{var(y_{predicted})}{var(y_{predicted}) + var(y_{residual})} \tag{7}$$

where $var(y_{predicted})$ is the variance of the set of predicted values by the linear regression model and $var(y_{residual})$ is the

variance of the set of residuals. We expect that wind-driven processes such as upwelling and mixing depend on cumulative rather than instantaneous wind stress. We thus perform a running back-average on the reference HRDPS wind stress record over the minimum averaging window required to maximize $R^2$.

**PCA mode attribution**

    In order to determine the importance of wind-driven upwelling as it relates to the PCA modes of variability for surface nitrate

and temperature, we must consider all of the relevant processes that contribute to this variability during the productive season. Given the regional attributes presented in Section 2.1, we expect these processes to be dominated by tides and diurnal heating and cooling at the daily time scale, and by wind and the fortnightly tidal cycle in the subtidal frequency range. We do not consider freshwater input since, although runoff is largest during the summer, this runoff is primarily driven by snow melt which we expect to vary slowly relative to our time scales of interest (Fleming and Clarke, 2005). We also disregard any

Fraser River plume variability which we expect to be primarily stochastic (Pawlowicz et al., 2019) or driven by wind and tides (Halverson and Pawlowicz, 2016). Likewise, we disregard the biological nitrate sink since drawdown events are tightly coupled to wind events (Del Bel Belluz et al., 2021) and we thus interpret the underlying process to be wind-driven. Given

the remaining processes of interest, we assign the dominant physical phenomena that can drive changes in surface nitrate and temperature during the productive season at frequencies higher than the 35 d high-pass filter cutoff into the following four categories: wind-driven upwelling, wind-driven mixing, tidal mixing and advection, and diurnal heating and cooling. For the purposes of this study, we acknowledge the presence of tidal mixing and advection at all time scales but we only explore the subtidal modulation of tidal mixing in detail since we are more interested in processes that overlap with the synoptic time scales described in Section 2.1.

We use the following four criteria for attributing specific surface nitrate and temperature PCA modes to these physical phenomena:

1. **Spatial anomalies** The EOF spatial pattern matches the expected surface tracer anomalies from a given process. Specifically, wind-driven upwelling produces coastal anomalies along the sides of the basin, tidal mixing produces anomalies in the tidal mixing regions, and wind-driven mixing and diurnal heating and cooling all produce uniform anomalies in the stratified areas of the basin.

2. **Power spectra** The power spectral density (PSD) of the PC loadings contains the expected features of the given process. Specifically, wind-driven upwelling and mixing both produce broadly distributed, red noise energy throughout the subtidal frequency range, tidal mixing produces significant energy peaks concentrated in the fortnightly and monthly subtidal bands, and diurnal heating and cooling produce significant energy peaks at the daily frequency.

3. **Wind coherence** The significance and frequency bands of the coherence between the PC loadings and the along-axis wind stress match the expected wind dependence associated with the given process. Specifically, wind-driven upwelling and mixing demonstrate significant coherence in the subtidal frequency range. Coherence is not significant for tidal mixing or diurnal heating and cooling.

4. **Wind correlation** The correlation between the PC loadings and the along-axis wind stress matches the expected wind dependence associated with the given process. Specifically, PCA modes attributed to wind mixing are correlated symmetrically with both positive and negative along-axis wind stress while PCA modes attributed to wind-driven upwelling are correlated asymmetrically with only one direction of wind stress. Correlation is insignificant for tidal mixing and diurnal heating and cooling.

## 2.4 Analysis software

SalishSeaCast results and HRDPS forcing data were stored and accessed in the Network Common Data Form (NetCDF) format (Unidata, 2019). All analysis and visualization was performed using Python 3.9 and the Jupyter development environment (Kluyver et al., 2016). In addition to the Python standard library, the following packages were used: Xarray for NetCDF processing (Hoyer and Hamman, 2017), NumPy for arrays and matrix operations (Harris et al., 2020), SciPy for filtering and spectral analysis (Virtanen et al., 2020), PyMC3 for Bayesian MCMC linear regression (Salvatier et al., 2016), Pandas

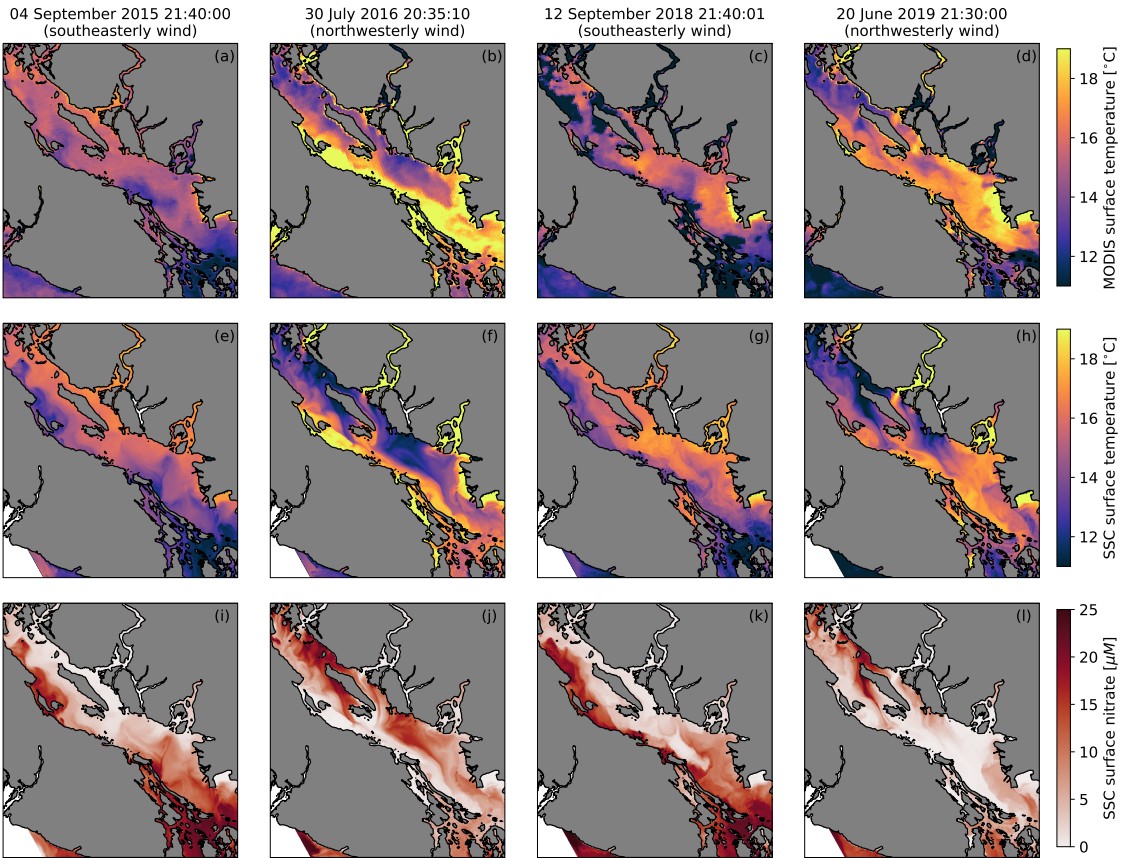

**Figure 2.** Surface temperature observations from the Moderate Resolution Imaging Spectroradiometer (MODIS) instrument aboard the NASA Aqua satellite (a-d), SalishSeaCast (SSC) surface temperature (e-h) and SSC surface nitrate (i-l) during four cloud-free upwelling events. All times are in UTC. The surface fields shown on 4 September 2015 and 12 September 2018 occurred during southeasterly wind and show cold, nitrate-rich upwelling along the Vancouver Island coast, while the surface fields shown on 30 July 2016 and 20 June 2019 occurred during northwesterly wind and show upwelling along the opposing BC mainland coast and the western side of Texada Island. The SSC fields reproduce the overall spatial structure observed in the MODIS images.

for loading and processing observed wind data (McKinney, 2010), Matplotlib for plotting (Hunter, 2007), Cartopy for map projections (Met Office, 2010-2015) and Windrose for polar histograms (Roubeyrie and Celles, 2018).

## 3 Results

### 3.1 Overview of 5 year record and comparisons with observations

In order to illustrate how SalishSeaCast resolves upwelling in the SoG, we first compare snapshots of hindcast surface temperature and nitrate with surface temperature observations from the Moderate Resolution Imaging Spectroradiometer (MODIS)

instrument aboard the NASA Aqua satellite. We selected four cloud-free images that overlap with sustained, along-axis wind events during the hindcast period. Two of these images were taken during southeast wind (4 September 2015 and 12 September

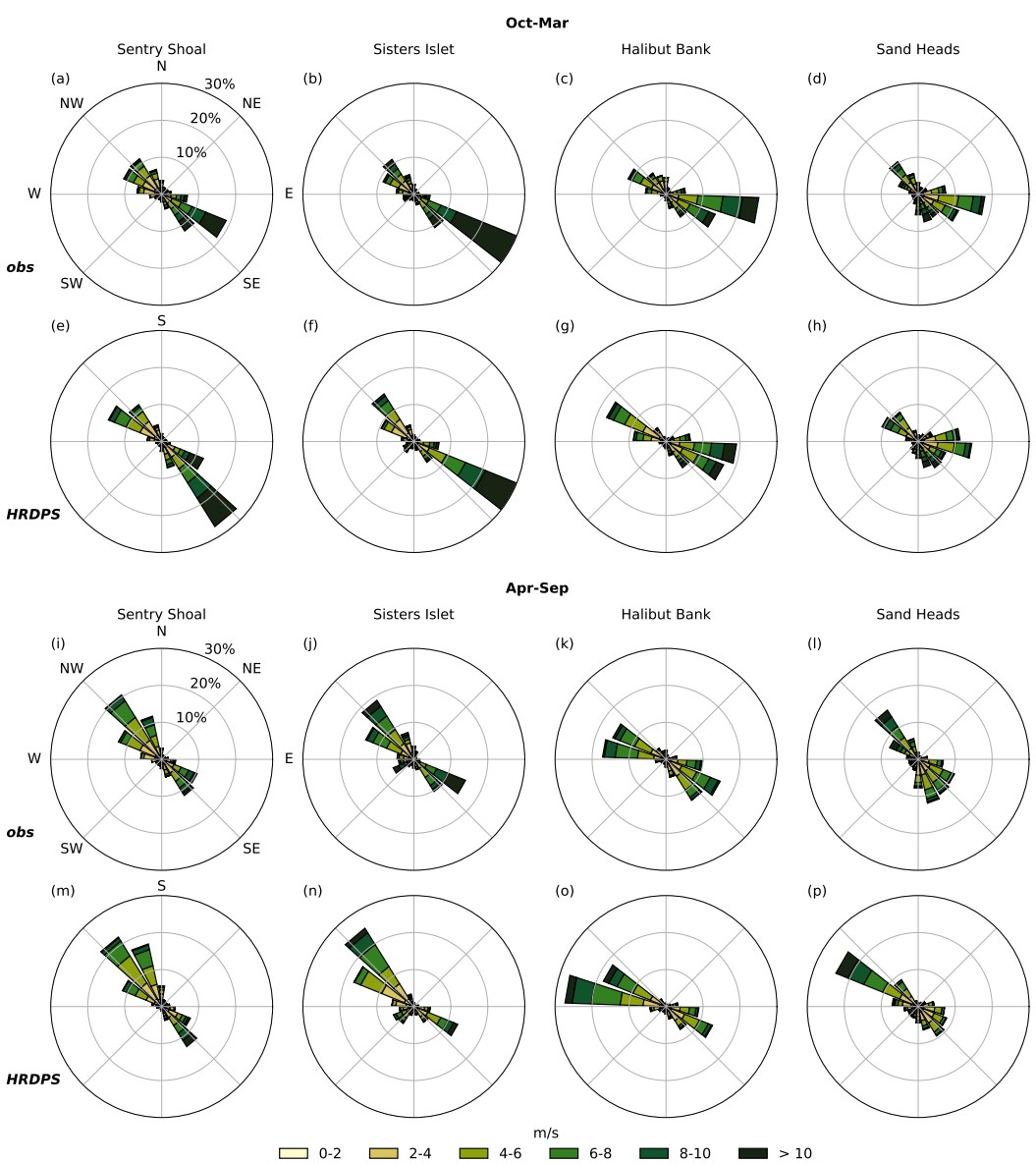

**Figure 3.** Condensed hourly wind station observations and hourly HRDPS results for the 2015 through 2019 period at Sentry Shoal (left column), Sisters Islet (second from left), Halibut Bank (second from right) and Sand Heads (right column). Data are binned by direction from and relative frequency of occurrence is given by the radial magnitude of each bin. Overall, the HRDPS wind record reproduces the observed, along-axis wind climatology at all stations. The seasonal wind separation of strong winter southeasterlies and weaker summer northwesterlies is also reproduced.

2018) and two were taken during northwest wind (30 July 2016 and 20 June 2019). In all four images, a cross-axis temperature gradient toward the right of the wind direction is clearly visible in the MODIS surface temperature observations (Fig. 2, top row), and the dominant spatial features of these gradients are reproduced in the SalishSeaCast surface temperature fields

(Fig. 2, middle row). These features include a cold coastal upwelling band to the left of the wind direction, offshore and downwind advection of the upwelling plume, and along-axis variability. The cold upwelling regions of the surface temperature fields also overlap with positive nitrate anomalies in the SalishSeaCast surface nitrate fields (Fig. 2, bottom row). From these four snapshots, upwelling appears to be consistently strong in specific areas and weak or patchy in others. Along the eastern shore, upwelling is strongest in the northern and central SoG and the northern tip of Texada Island, and weakens significantly

in the southern SoG near the Fraser River mouth. Along the western shore, upwelling is most prominent in the northern SoG near Baynes Sound. In addition, large nitrate plumes appear to be advected northward from the Haro Strait region where tidal mixing is strong.

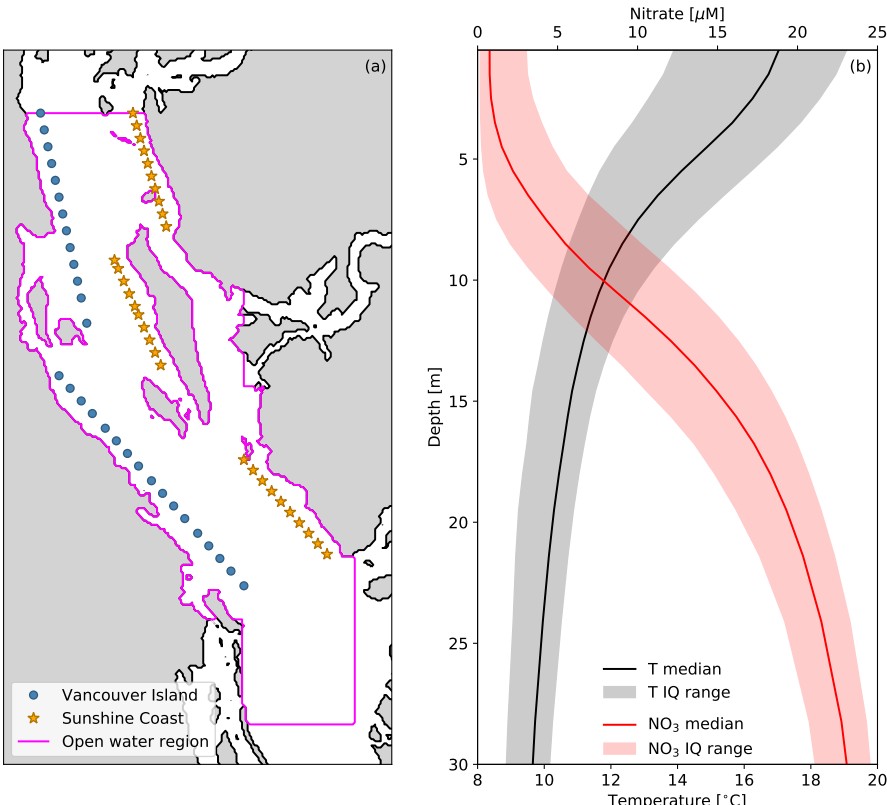

**Figure 4.** (a) Locations of grid points and regions used to obtain the spatial median time series of hourly temperature, nitrate and HRDPS along-axis wind speed shown in Fig. 5. (b) Time-median depth profiles of SalishSeaCast nitrate (NO$_3$, red line) and temperature (T, black line) during the productive seasons of 2015-2019 over the magenta open water region shown in (a). The seasonal cutoffs for each year are shown in Fig. 5 and the interquartile (IQ) range across all seasons is also shown (shaded envelopes).

We next compare the 5 year HRDPS hourly surface wind velocity record with 5 years of hourly wind velocity observations at four open water, meteorological observing stations shown in Fig. 1 (magenta symbols). Sand Heads and Sisters Islet are

navigational light stations equipped with meteorological sensors maintained by ECCC, while Halibut Bank and Sentry Shoal are 3 m discus buoys maintained by the Department of Fisheries and Oceans Canada (DFO). At all four stations, an along-axis wind dominance is clear from the observed wind velocity record in winter (Fig. 3, top row) and summer (Fig. 3, second row from bottom), and the primary features of this record are reproduced by the HRDPS wind velocities (Fig. 3, second row from top and bottom row). In addition to along-axis dominance, these features include a seasonal separation between strong

southeasterly winter wind and weaker northwesterly summer wind. Winter southeasterly winds are stronger in the northern SoG relative to the southern SoG, however the summer northwesterlies do not show an along-axis variation as clearly. The along-axis dominance and seasonal directionality of these records is consistent with the synoptic groupings and topographic steering discussed in Sec. 2.1 and provides a potentially significant physical driver for upwelling in the SoG.

With the along-axis wind dominance and seasonality established, we now turn our focus to the 5 year SalishSeaCast nitrate

and temperature records. We first examine these records as spatial medians over an open water region of the domain and over two sets of grid points along the eastern Sunshine Coast and western Vancouver Island coast (Fig. 4a). Applying first a time median over the productive season to profiles of nitrate and temperature in the open water region, we demonstrate the seasonal vertical gradients, nitracline and thermocline respectively, that form in spring and erode in autumn (Fig. 4b). The nitracline is centered at approximately 5-15 m depth while the thermocline is shifted closer to the surface between 0 and 10 m depth.

The small interquartile range for surface nitrate is consistent with continuous biological drawdown while the large interquartile range for surface temperature is indicative of more frequent sources of variability like diurnal heating and wind mixing.

Looking next at the surface, time series of median, open water surface nitrate and temperature exhibit seasonal cycles that follow the expected cycles of wind, solar and freshwater forcing described in Sec. 2.1 (Fig. 5, top panel). During winter when we anticipate elevated storm activity, surface nitrate is high and surface temperature is low. During summer when we anticipate

strong stratification and biological nutrient uptake, surface nitrate decreases and surface temperature increases. The seasonality of wind speed and direction demonstrated in Fig. 3 is reflected in the spatial median of along-axis HRDPS wind speed over the same open water region (Fig. 5, top panel, gray patch). Strong positive, or southeasterly, wind events dominate during the winter while weaker negative, or northwesterly, wind events dominate during the summer.

During the productive season, the spatial medians of surface nitrate and temperature along Vancouver Island and the Sunshine

Coast exhibit significant episodic variations that overlap with corresponding variations in the along-axis wind record (Fig. 5, lower panels). The nitrate variability is clearly asymmetric between the two coasts, where variations along Vancouver Island are stronger during positive, or southeasterly, wind events while variations along the Sunshine Coast are stronger during negative, or northwesterly, wind events (Fig. 5, left panels). The episodic temperature variability is less clearly attributed to specific wind events or a given wind direction, and significant seasonal variability is still present within the productive season cutoffs (Fig. 5,

right panels). However, the four MODIS surface temperature images and SalishSeaCast surface tracer snapshots presented in Fig. 2 overlap clearly with well-defined nitrate and temperature variations along the Vancouver Island coast in September 2015 and 2018 during southeasterly wind, and along the Sunshine Coast in July 2016 and June 2019 during northwesterly wind

(Fig. 5, green lines). Large interquartile ranges along each coastline, especially in nitrate (Fig. S1), suggest that significant along-shore variability is generally present during surface tracer anomaly events like those shown in Fig. 2.

## 345  3.2  Principal component analysis

When principal component analysis is applied to the de-trended surface nitrate and temperature records during the productive season resampled over the SoG subdomain shown in Fig. 1, only the first three nitrate modes and the first temperature mode

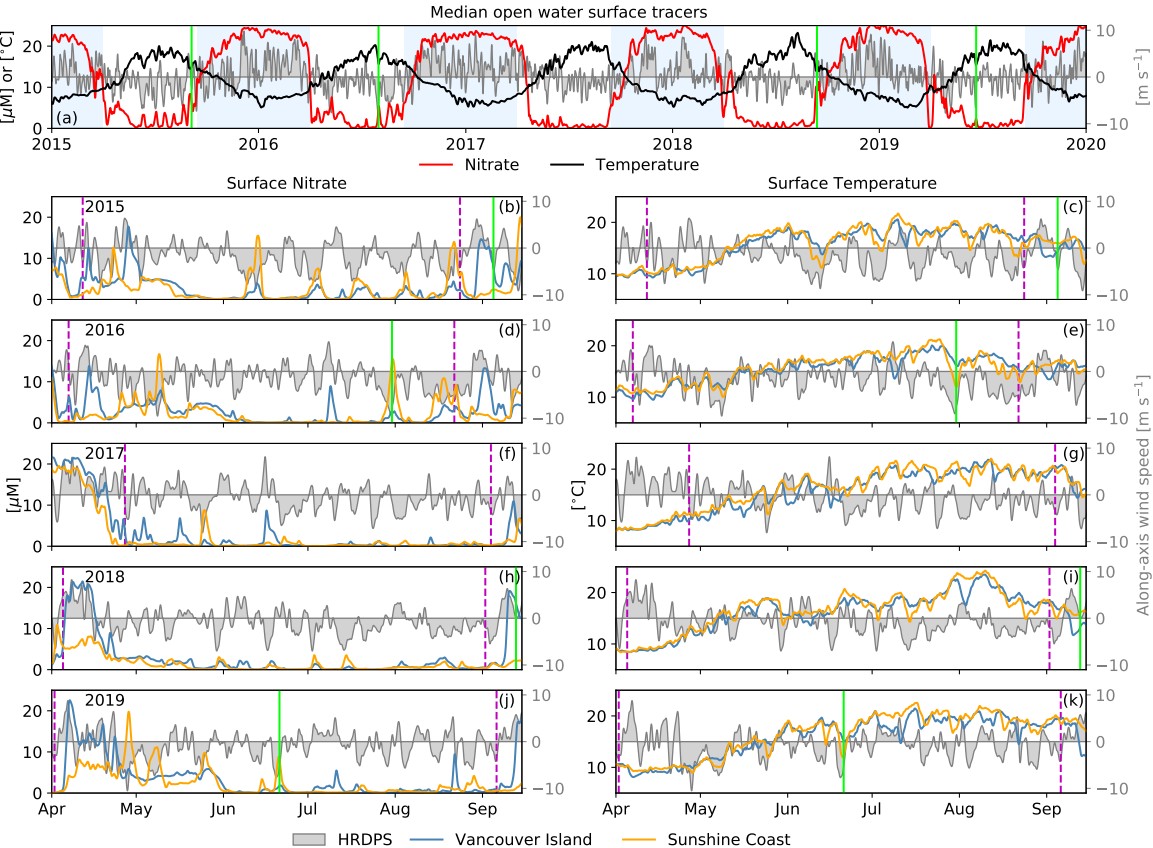

**Figure 5.** Time series of SalishSeaCast hourly surface nitrate and temperature between 2015-2019 as (a) spatial medians over the open water region and (b-k) spatial medians over the coastal grid points shown in Fig. 4. The open water spatial median of HRDPS hourly along-axis surface wind speed is also shown (gray patch). Interquartile ranges for temperature and nitrate are included in a companion figure in the supplement (Fig. S1) The nitrate, temperature and wind speed records have been low-pass filtered using a 3-day Blackman window to emphasize the subtidal variability. The 1 April to 15 September extent of the lower panels is indicated by the white regions in (a). The seasonal cutoffs bounding the productive season for each year (dashed magenta lines) and the dates of the four MODIS images shown in Fig. 2 (green solid lines) are also shown. Nitrate and temperature follow clear seasonal cycles, with higher frequency variability during the summer that visibly overlaps with wind variability.

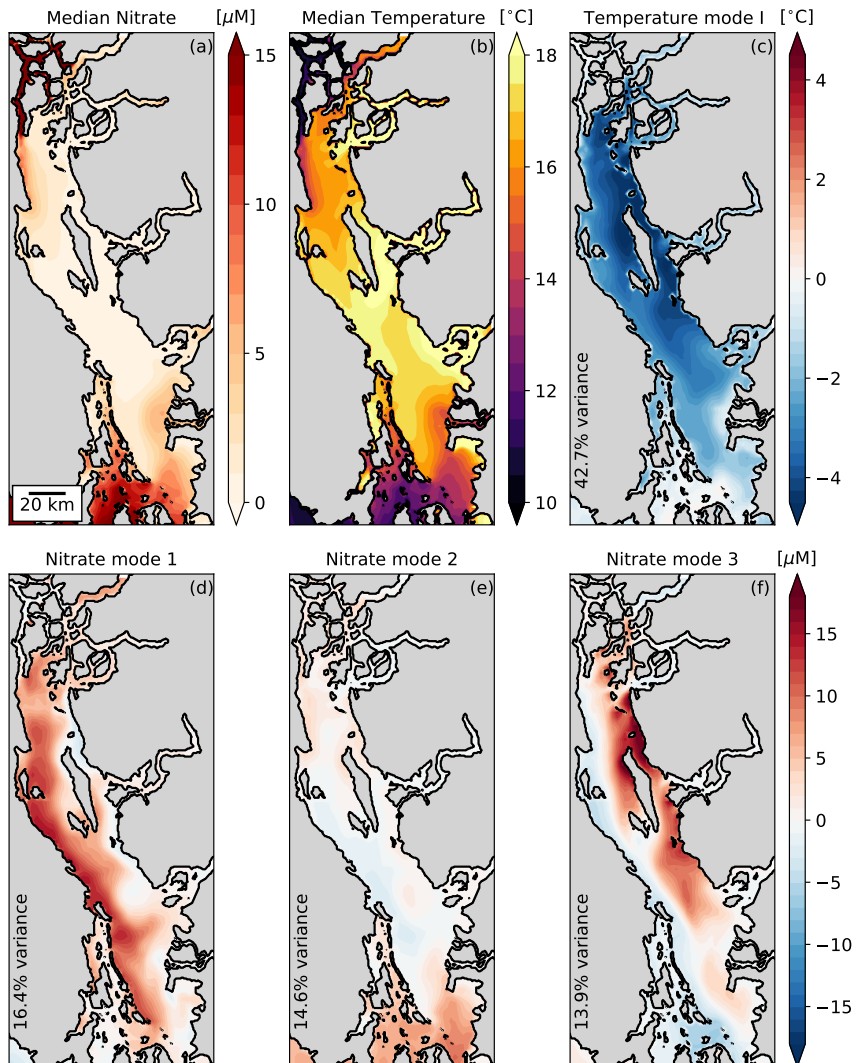

**Figure 6.** Empirical orthogonal function (EOF) spatial anomalies in order of decreasing percent variance applied to the 2015-2019 Salish-SeaCast surface temperature (c) and nitrate (d-e) records between the seasonal cutoffs shown in Fig. 5. The median surface nitrate (a) and temperature (b) fields between the seasonal cutoffs are also shown. The surface fields are subsampled to 2.5 km resolution and detrended using a 35 d high-pass Blackman window filter prior to the principal component analysis (Section 2.3). The variance explained by each mode is printed in the lower left corner of each panel. Nearly half of the total variance is distributed approximately evenly across the first three nitrate modes (d-f) while a similar 42.7% of temperature variance is contained entirely within the first temperature mode (c).

account for greater than 10% each of the total variance. Nearly half of the variance for each tracer is explained by these modes (44.9% and 42.7% respectively for nitrate and temperature), with approximately equal variance distribution across the three

nitrate modes. For the remainder of this analysis, we reference these modes as nitrate modes 1, 2 and 3 and temperature mode I, and we disregard the higher modes that account for increasingly small fractions of the total variance.

Following the mode attribution criteria described in Section 2.3, we first examine the EOF spatial anomaly patterns for each mode (Fig. 6, S2). The median surface nitrate and temperature fields are included to show the background state of each tracer (Fig. 6a and b). As expected from the median profiles shown in Fig. 4b, surface nitrate is depleted and surface temperature

is high in the interior SoG, with each tracer respectively increasing and decreasing approaching the tidal mixing zones to the north and south. The nitrate mode 1 and 3 EOF patterns show clear coastal anomalies in the nitrate-depleted interior SoG, indicative of upwelling along the respective western and eastern coastlines. Conversely, the nitrate mode 2 EOF pattern shows no significant anomalies in the interior SoG, but instead an anomaly in the southern passages near Haro Strait indicative of fortnightly tidal mixing modulation. The temperature mode I EOF shows a uniform anomaly in the interior SoG that vanishes

toward the tidal mixing regions. Based on our mode attribution criterion (1), this uniform anomaly is not consistent with upwelling or fortnightly modulation of tidal mixing strength and is instead indicative of either wind mixing or diurnal heating and cooling.

We next examine the PC loadings time series for each mode, beginning with the primary upwelling candidates, nitrate modes 1 and 3 (Fig. 7). Both modes feature pronounced positive anomalies during wind events similarly to the medians along

the coastal transects shown in Fig. 5. Nitrate mode 1 increases during sustained positive, southeasterly wind while nitrate mode 3 increases during sustained negative, northwesterly wind. Nitrate modes 1 and 3 are not completely isolated in their wind responses, however. The mode 1 PC, for example, also weakly increases during northwesterly wind, which we also observed in the spatial median along Vancouver Island in Fig. 5.

Looking next at the power spectral density (PSD) of the PC loadings for each mode, the nitrate mode 1 and 3 and temperature

mode I power spectra all demonstrate auto-regressive, red noise behavior with broadly distributed energy in the subtidal range, while the nitrate mode 2 PSD is more focused with significant peaks at the fortnightly and monthly frequencies (Fig. 8a). These power spectra suggest that nitrate modes 1 and 3 and temperature mode I are wind-driven while nitrate mode 2 is driven by the fortnightly tidal cycle. The nitrate mode 3 PSD also shows a significant peak in the 15-20 d band (Fig. 8a, orange curve) that is consistent with the visual spacing of the mode 3 anomalies in Fig. 7. All four modes have significant peaks in the tidal

range, indicating that tidal energy is never fully separated by the PCA method even when varimax rotation is used. We make no further distinctions about PSD in the tidal range, with the exception of the prominent diurnal peak in the temperature mode I spectrum (Fig. 8a, black curve) which is indicative of diurnal heating and cooling.

All four PCA modes demonstrate some degree of significant spectral coherence with along-axis wind stress beyond the 99% confidence interval (Fig. 8b). Nitrate mode 3 is significantly coherent with wind stress across the subtidal frequency range with

maximum coherence in the 15-20 d band, while the remaining modes are coherent only at isolated frequencies within the 2.5-10 d band. The increased coherence at lower frequencies for nitrate mode 3 is consistent with the visual spacing of the nitrate mode 3 PC anomalies in Fig. 7 and with the longer maximum duration of summer synoptic types mentioned in Section 2.1. Conversely, the higher frequency coherence bands for nitrate mode 1 are more consistent with the shorter maximum duration of winter synoptic types. Additionally, we have likely selected for the higher overall coherence observed in nitrate mode 3 by

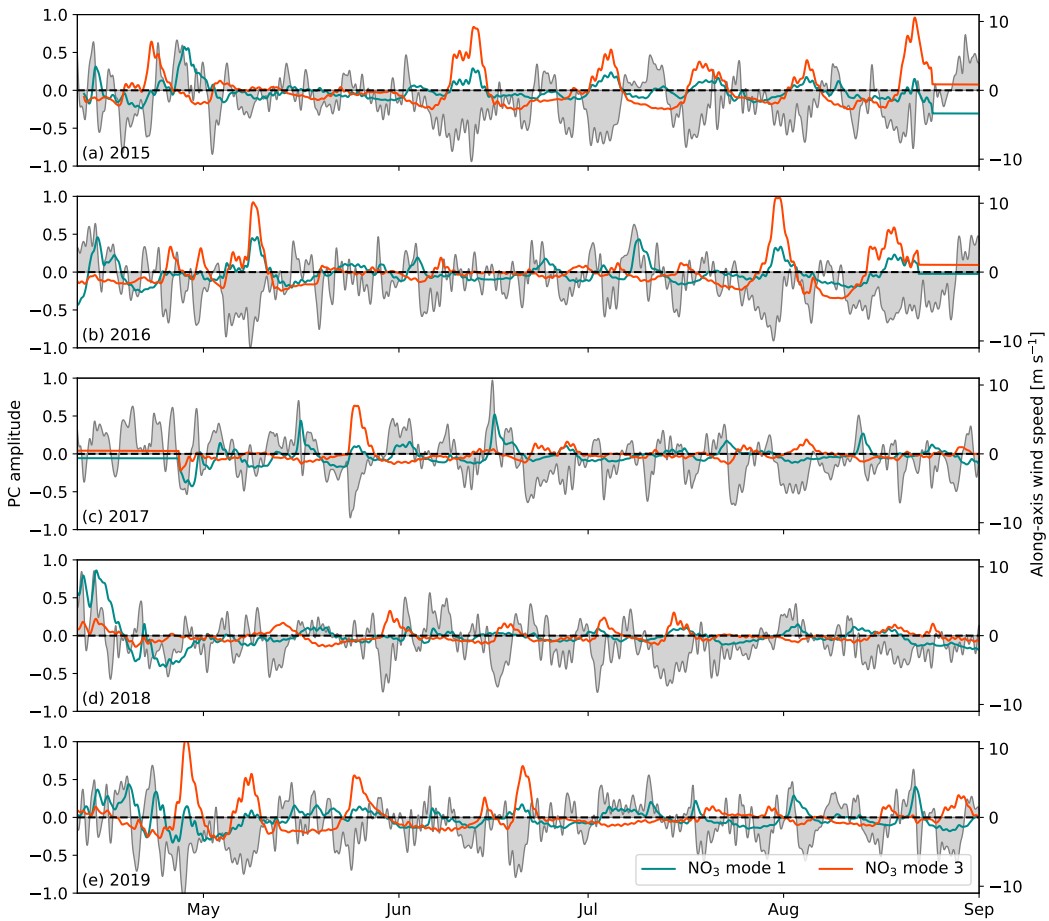

**Figure 7.** Time series of the principal component (PC) loadings for surface nitrate modes 1 and 3 corresponding to the EOF spatial patterns shown in Fig. 6. The spatial median of along-axis HRDPS wind speed over the open water region in Fig. 4a (magenta region) is also shown. The PC and wind time series are low-pass filtered using a 1 d Blackman window cutoff in order to emphasize the low frequency variability. The PC loadings of both nitrate modes increase during sustained wind events.

restricting our analysis to the productive season when northwesterly wind dominates. Nitrate mode 2 demonstrates significant coherence in the 3-10 d band, however since the mode 2 PSD is concentrated in the fortnightly and monthly bands, we do not associate this coherence with significant PC variability. Temperature mode I demonstrates the weakest coherence of all modes, which may simply be the result of a complicated response to both wind and solar forcing.

The time lag between a wind event and a corresponding PC anomaly can be estimated from the phase lag associated with
the coherence calculation (Fig. 8c). Nitrate mode 2 is not shown since we have disregarded the significance of the coherence. Likewise, we have limited the nitrate mode 1 and temperature mode I lag estimates to the 2.5-9 d frequency range since coherence for these modes is not significant outside of this range. We do not attempt to draw meaningful conclusions about

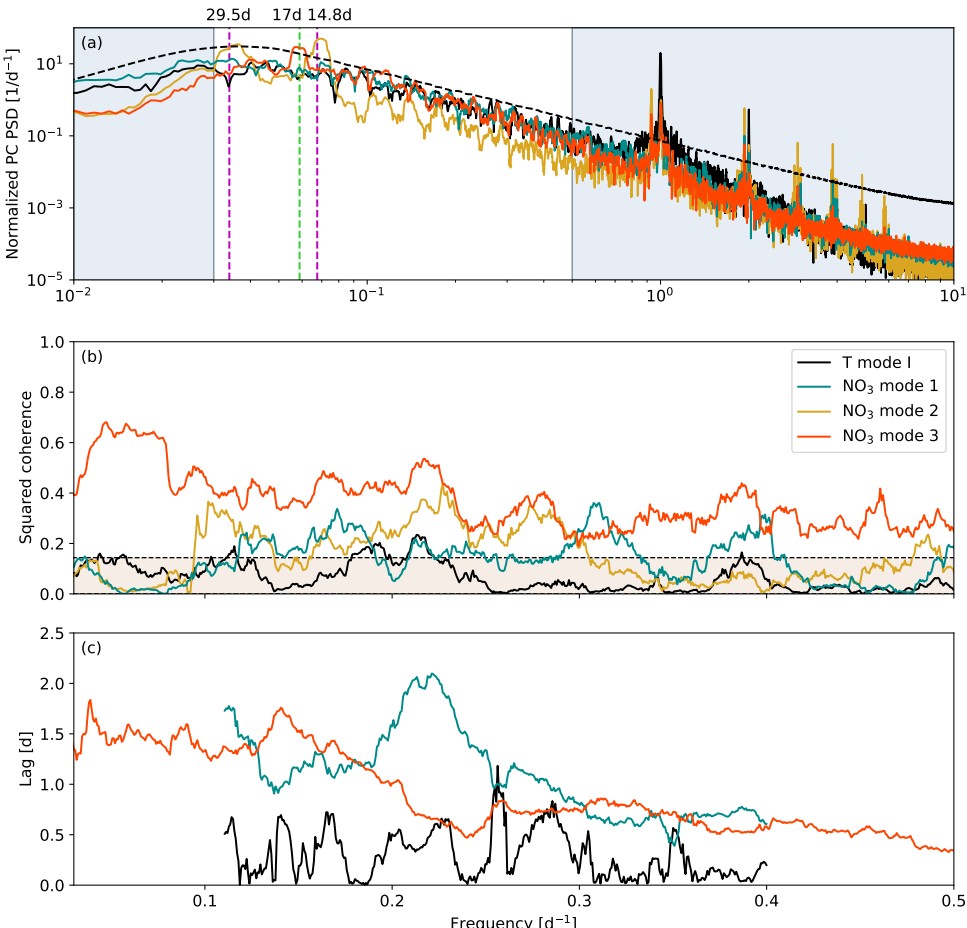

**Figure 8.** (a) Normalized power spectral density (PSD) and (b) spectral coherence with along-axis, HRDPS wind stress calculated from the principal component (PC) loadings of the four surface tracer PCA modes shown in Fig. 6. The phase lag associated with the coherence calculation is shown as a peak-to-peak time lag (c) for nitrate modes 1 and 3 and temperature mode I. The fortnightly and monthly subtidal frequencies are shown in (a) (magenta dashed lines), and a subtidal peak in nitrate mode 3 centered at 17 d is also shown (green dashed line). The periods associated with these three dashed lines are printed at the top of panel (a). Panels (b) and (c) are shown on a linear frequency axis over the non-shaded region in (a). The 99% confidence intervals for PSD and coherence are shown by the black dashed lines in (a) and (b). Peak-to-peak lags for nitrate mode 1 and temperature mode I are only shown in the 2.5-9 d band since neither mode is significantly coherent with wind stress beyond this range.

the lag variability or frequency distribution. Instead we simply observe that the lags associated with nitrate modes 1 and 3 are similar and range between approximately 0.5 and 2 d while the lag associated with temperature mode I is shorter, ranging from 0-1 d. These estimates reflect the peak-to-peak lag between signals and thus do not account for the onset of a wind event which precedes the peak.

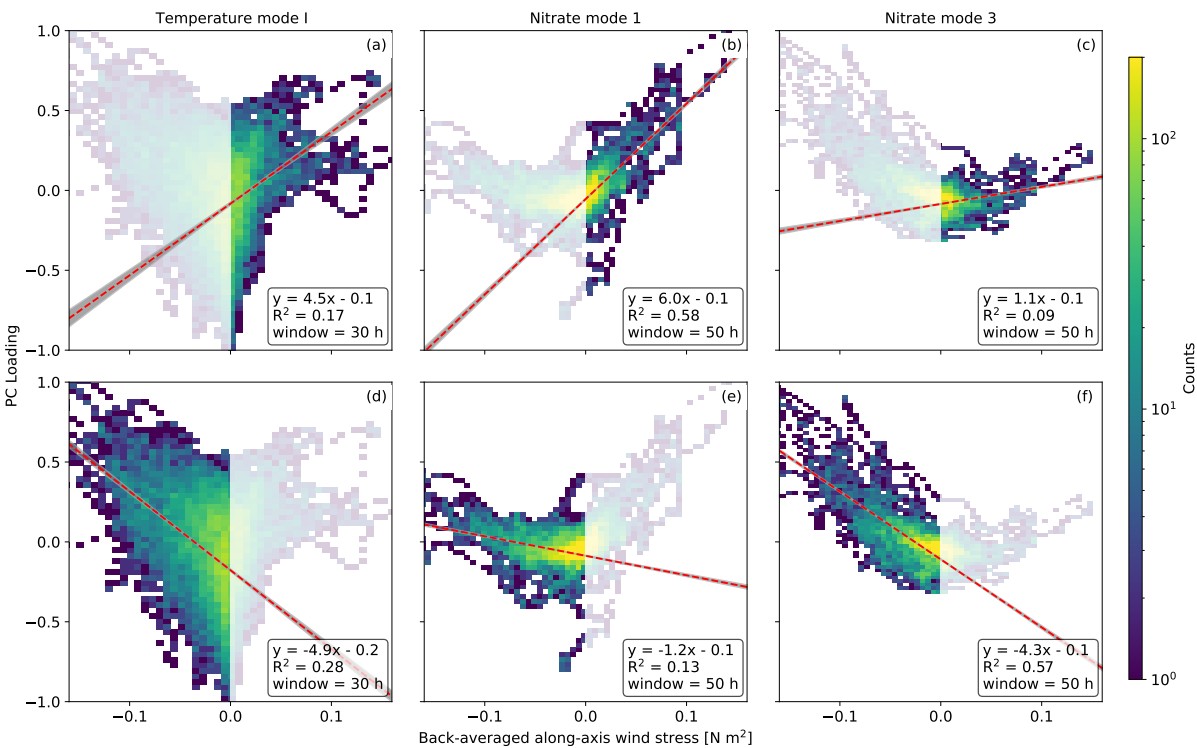

**Figure 9.** 2-D histograms of the PC loadings for temperature mode I (a, d), nitrate mode 1 (b, e) and nitrate mode 3 (c, f) versus the temporally back-averaged spatial median of along-axis wind stress calculated from the HRDPS wind record according to Equation 2. Bayesian linear regression is performed separately for positive, southeasterly (a-c) and negative, northwesterly (d-f) wind stress. Regression statistics shown include a mean best fit (red line and equation), a 99% confidence interval (gray envelope), a correlation coefficient ($R^2$) and a wind stress back-averaging window tuned to maximize $R^2$. The temperature mode I correlation is strongly symmetric between positive and negative wind stress, and the back-averaging window is approximately 30 h for both directions. Conversely, the nitrate modes 1 and 3 correlations are strongly asymmetric between wind stress directions, and the back-averaging window is thus determined according to the wind stress direction that yields the highest regression slope and $R^2$ value. This direction is positive for nitrate mode 1 (b) and negative for nitrate mode 3 (f), with a 50 h averaging window for both modes.

The final mode attribution criterion described in Section 2.3 is the correlation between the PC loadings and the back-averaged along-axis wind stress, shown for the wind-forcing candidates, nitrate modes 1 and 3 and temperature mode I, in Fig. 9. All three modes increase with wind stress in both along-axis directions. We therefore calculate linear regression and the correlation coefficient $R^2$ for each mode separately for positive, southeasterly wind stress (Fig. 9a-c), and for negative, northwesterly wind stress (Fig. 9d-f). The minimum back-averaging window required to maximize the correlation between temperature mode I and either direction of wind stress is approximately 30 h. Nitrate modes 1 and 3 are more asymmetric in their correlation with wind stress, and we determine the minimum back-averaging window based on the sign of wind-stress that yields the stronger correlation. This sign is positive for nitrate mode 1 and negative for nitrate mode 3, with a window of approximately 50 h required to maximize the correlation for both nitrate modes. This longer window relative to the 30 h window determined for the temperature mode is consistent with the peak-to-peak lags that we observed in Fig. 8c.

Linear regressions for all modes against both positive and negative wind stress yield significant correlations, in that the probability distribution of each regression slope never contains zero within the 99% confidence interval (Fig. 9, grey envelopes). This significance combined with the EOF patterns, power spectra and coherence satisfies our criteria for attributing wind as a dominant forcing of these modes. The mean regression slopes and correlation coefficients provide a further basis for quantifying and interpreting the asymmetry between the southeasterly and northwesterly wind stress dependence of each mode. Nitrate mode 1 is strongly asymmetric with a significantly higher regression slope and $R^2$ for positive, southeasterly wind stress and nitrate mode 3 is similarly asymmetric but with stronger correlation statistics for negative, northwesterly wind stress. Conversely, temperature mode I demonstrates weaker asymmetry, with a marginally stronger correlation during negative, northwesterly wind.

Reviewing the mode attribution criteria described in Section 2.3, we attribute the following physical processes to each PCA mode. Based on the coastal nitrate mode 1 and 3 EOF anomalies, the red-noise distribution of subtidal spectral energy, the significant coherence with along-axis wind stress in the expected frequency ranges and the asymmetric correlation between positive and negative wind stress, we attribute nitrate modes 1 and 3 to upwelling along the respective western and eastern coastlines. Conversely, given the nitrate mode 2 EOF anomaly in the southern tidal mixing region, the significant PSD peaks at the fortnightly and monthly frequencies, the lack of significant wind-stress coherence at these frequencies and the lack of wind stress correlation (not shown), we attribute nitrate mode 2 to changes in tidal mixing strength due to the fortnightly tidal cycle. Finally, based on the uniform temperature mode I EOF anomaly in the interior SoG, the red-noise subtidal spectral energy distribution coupled with a pronounced diurnal peak and the symmetric correlation between positive and negative wind stress, we attribute temperature mode I to a combination of wind mixing and diurnal heating and cooling. Given these mode attributions, we conclude that subtidal surface nitrate variability during the productive season is dominated by upwelling and fortnightly tidal modulation while subtidal surface temperature variability is dominated by wind mixing and diurnal heating and cooling. Since tidal mixing anomalies are limited to the ends of the basin, we further conclude that upwelling is the dominant source of surface nitrate to the interior SoG during the productive season.

 **4 Discussion**

## 4.1 Upwelling mechanisms

To provide a mechanistic context to the cross-axis upwelling response observed in nitrate PCA modes 1 and 3, we compare our results with two-layer shallow water theory in a closed basin. We consider a two-layer model to be sufficiently suited for describing upwelling across the pycnocline despite the established three-layer structure of the SoG since the deep layer is
generally observed to be below 200 m depth (Pawlowicz et al., 2007). We also disregard the complicating factors of the stratified surface "mixing layer" (e.g., Collins et al., 2009) since the onset of sustained wind homogenizes this layer in SalishSeaCast in a matter of hours (simulations not shown) compared with the longer upwelling response that we determined during the mode attribution on the order of days. Additionally, upwelling solutions of two-layer problems retain many of their defining characteristics as additional layers are added (Csanady, 1982a).

**Two layer model description**

Consider a wind forced, two layer, linearized, shallow water approximation of a rectangular basin as described by Csanady (1982b). Assuming that baroclinic motions dominate the momentum balance, the lower layer transports are equal and opposite to the upper layer transports and the interface displacement $\zeta$ is coupled to the surface displacement $\eta$ according to

$$\eta = -\frac{\Delta\rho}{\rho_0}\frac{h_1 + h_2}{h_2}\zeta \tag{8}$$

where $\Delta\rho$ is the density difference across the pycnocline, $\rho_0$ is the background density, and $h_1$ and $h_2$ are the respective upper and lower layer thicknesses. This coupling between the upper and lower layers results in a single set of momentum and continuity balances for the upper layer cross-axis and along-axis transports, $U$ and $V$, respectively

$$\frac{\partial U}{\partial t} - fV = g'h_1\delta_h\frac{\partial\zeta}{\partial x} \tag{9a}$$

$$\frac{\partial V}{\partial t} + fU = g'h_1\delta_h\frac{\partial\zeta}{\partial y} + \delta_h\frac{\tau}{\rho_0} \tag{9b}$$

$$\frac{\partial U}{\partial x} + \frac{\partial V}{\partial y} = \frac{\partial\zeta}{\partial t} \tag{9c}$$

where $x$, $y$ and $t$ are the respective cross-axis, along-axis and time coordinates, $f$ is the Coriolis parameter, $g' = g\Delta\rho/\rho_0$ is the reduced gravitational acceleration across the interface, $g$ is the gravitational acceleration constant and $\tau$ is the along-axis wind stress. The thickness ratio $\delta_h = h_2/(h_1 + h_2)$ determines the amount of coupling between the surface forcing $\tau$ and the lower layer motions. For a deep SoG, assume $h_1/h_2 << 1$ and $\delta_h \approx 1$. For the moment, we assume negligible bottom friction.
The balance between the wind stress term and the pressure gradient force term is a recurring element in solutions to Equation 9. Assuming that the shear stress between layers translates directly from the surface wind stress, which is common in two-layer models of lakes (e.g., Stevens and Lawrence, 1997; Stevens and Imberger, 1996), this balance can be summarized by a bulk Richardson number $Ri = g'h_1\rho_0/\tau$. Additionally, there are three fundamental length scales to consider: the along-axis length $L_A$, the cross-axis length $L_C$ and the internal Rossby deformation radius $L_R$, which is redefined from Equation 1

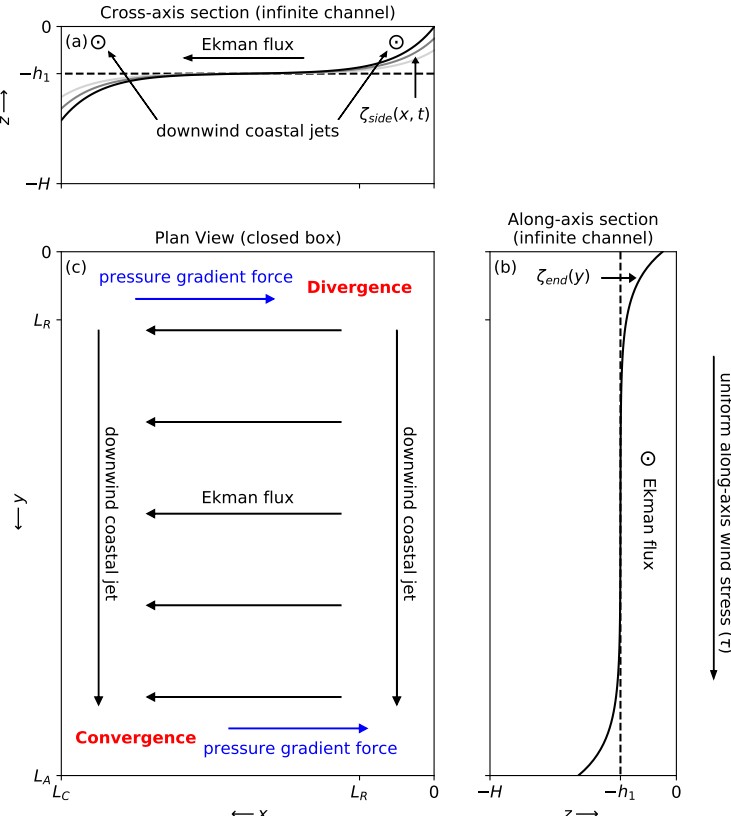

**Figure 10.** Diagram of the two-layer, infinite channel, interface displacement solutions in the (a) cross-axis and (b) along-axis directions, as forced by a uniform along-axis wind stress. The plan view (c) shows the surface divergence and convergence that are reasoned to occur when considering the infinite channel solutions in a closed basin.

for our two-layer approximation as $L_R = \sqrt{g'h_1}/f$. Near a coastline, $L_R$ represents the transition between pressure gradient dominance in the near-shore region and Ekman flux dominance in the offshore region. Naturally, if $L_C/L_R << 1$ then pressure gradients dominate everywhere and steady Ekman fluxes cannot develop.

Here we explore two well-established solutions to Equations 9a-c in the context of a northerly wind event that drives upwelling along the eastern shore. These solutions both assume an infinite channel with no gradients parallel to the channel walls.

Once the channel solutions are established, we will consider the effects of closing the basin. We orient our coordinates such that $x$ increases to the west, $y$ increases to the south, the northeast corner is located at $x = 0, y = 0$ and $\tau$ is positive southward (Fig. 10).

**Infinite channel solutions**

Consider first the coasts along the sides of the basin at $x = 0$ and $x = L_C$ and neglect all along-axis gradients. After an initial spin-up period over which the cross-shore acceleration $\partial U/\partial t \to 0$, the surface wind stress drives growing along-shore jets near the coastal $U = 0$ boundary conditions and a steady cross-shore Ekman flux far from the coasts where $\partial \zeta/\partial x \to 0$. Since the growing along-shore jets are in geostrophic balance with the cross-shore pressure gradient force, the interface tilt also grows with time. Assuming steady $U$ and a linear time dependence for $V$, an approximate solution for $\zeta$ is presented in Csanady (1973). The upwelling solution along the eastern shore is

$$\zeta_{\text{side}}(x,t) = \frac{L_{\text{R}}}{Ri} f t e^{-x/L_{\text{R}}} \tag{10}$$

while a reversed downwelling solution arises along the western shore (Fig. 10a). The interface displacement at the coast is thus a linear function of $ft$ determined by $Ri$ and $L_{\text{R}}$.

Next consider the coasts at the upwind and downwind ends of the basin at $y = 0$ and $y = L_A$ and neglect all cross-axis gradients. After an initial spin-up period over which all accelerations $\partial U/\partial t \to 0$ and $\partial V/\partial t \to 0$, the surface wind stress balances a steady pressure gradient force near the end coasts and a steady cross-wind Ekman flux far from the end coasts where $\partial \zeta/\partial y \to 0$. The interface displacement only grows large enough to accommodate this pressure gradient force balance. Assuming steady $U$ and zero $V$, an approximate solution for $\zeta$ is also presented in Csanady (1973). The upwelling solution at the upwind end is

$$\zeta_{\text{end}}(y) = \frac{L_{\text{R}}}{Ri} e^{-y/L_{\text{R}}} \tag{11}$$

while a reversed downwelling solution arises along the downwind end (Fig. 10b). The interface displacement at the coast is once again determined by $Ri$ and $L_{\text{R}}$ but the solution is now steady with no continued growth. Comparing Equations 10 and 11, $\zeta_{\text{side}}$ clearly exceeds $\zeta_{\text{end}}$ for wind impulse durations longer than $t = 1/f$. Since $1/f \approx 2.5$ h at the SoG latitude while the time scales associated with the peak-to-peak lags and wind stress back-averaging windows are significantly longer, $\zeta_{\text{end}}$ is unlikely to contribute significantly to upwelling.

While the linear time approximation used to obtain $\zeta_{\text{side}}$ becomes increasingly invalid as upwelling progresses, the interface displacement presumably continues to grow by some unknown function of $t$ until the coastal jet is balanced by bottom friction. This frictional adjustment period can be estimated by adding a quadratic bottom friction term to the lower layer counterpart of Equation 9b at the coast

$$\frac{\partial V_2}{\partial t} = -\frac{\tau}{\rho_0} - C_{\text{b}} \left(\frac{V_2}{h_2}\right)^2 \tag{12}$$

where $V_2$ is the lower layer along-axis transport and $C_{\text{b}}$ is the bottom drag coefficient. The Coriolis term has vanished due to the coastal boundary condition and the along-axis pressure gradient has been neglected. A solution for $V_2$ is presented by Csanady (1974) and decays with time over an $e$-folding time scale of $T_F = h_2/\sqrt{4C_b\tau/\rho_0}$. For a typical drag coefficient of $C_{\text{b}} = 2 \times 10^{-3}$ and a nominal lower layer thickness of $h_2 = 100$ m, a wind stress of $\tau = 0.2$ N m$^{-2}$ corresponding to a wind

impulse of approximately 10 m s$^{-1}$ produces a $T_F$ on the order of 1 d. This estimated frictional adjustment period is again smaller than the upwelling time scales suggested by our PCA results, however $T_F$ is still approximately an order of magnitude larger than the inertial time scale that governs $\zeta_{\text{end}}$.

**Effects of the closed basin**

The infinite channel approximations ignore fundamental mass conservation discontinuities at the ends of the basin where $x, y = 0$ and $x, y = (L_C, L_A)$. The downwind coastal jets along the sides of the basin combined with the cross-wind Ekman flux in the basin interior result in surface divergence and convergence at the respective upwind and downwind corners when the domain becomes closed (Fig. 10c). While simple analytical solutions that resolve these discontinuities are difficult to obtain, we can reason that these divergences will ultimately cause the interface to shoal at $x, y = 0$ and deepen at $x, y = (L_C, L_A)$ beyond what is described in the channel solutions.

Eventually, the surface tilts that accompany these interface displacements (Equation 8) will produce cross-axis pressure gradient forces that oppose the cross-axis Ekman fluxes near the ends of the basin (Fig. 10c, blue arrows), allowing downwind surface transport in the basin interior. As these cross-axis pressure gradients continue to develop along the entire length of the basin, they will eventually shut down all cross-axis Ekman fluxes. At this point, the along axis pressure gradient force will be the only term remaining to balance the wind stress once steady state is reached, and the resulting along-axis interface tilt can be described by the familiar wind setup equation

$$\zeta_{\text{setup}}(y) = \frac{L_A/2 - y}{Ri} \tag{13}$$

The interface displacement at the basin ends is now determined by $L_A/2Ri$. Comparing this lake setup to the cross-axis channel solution, $\zeta_{\text{side}} = \zeta_{\text{setup}}$ when $t = L_A/2fL_R$. In the SoG, this required time is of similar order to the frictional time scale estimated above, which suggests that, if the lake setup were ever fully achieved, then $\zeta_{\text{setup}}$ and $\zeta_{\text{side}}$ would be of similar order as well.

The scenario we have presented to account for this along-axis, lake setup in the SoG relies on the basin being completely closed. In reality, the SoG is not closed but is instead connected to neighboring basins by complicated networks of passages. Given the presence of these passages, $\zeta_{\text{setup}}$ is unlikely to contribute significantly to upwelling relative to $\zeta_{\text{side}}$, and the cross-axis upwelling described by $\zeta_{\text{side}}$ is ultimately what we observe in the raw hindcast surface tracers and in the PCA results. One major exception to this interpretation is the along-axis gradient in the nitrate PCA mode 3 coastal anomaly, which indicates that nitrate upwelling along the Sunshine Coast is consistently stronger in the northern SoG relative to the southern SoG. This regional upwelling pattern is consistent with at least a partial development of the corner divergence model in Fig. 10c. However, as we will discuss, there are other regional considerations that may explain this upwelling pattern as well.

**4.2  Other geographic considerations**

In Section 4.1, we suggested that the along-axis surface nitrate anomaly gradient in the PCA mode 3 EOF pattern (Fig. 6f) could be the result of a partial development of the corner divergences that we proposed to occur in a completely closed

basin. However, this anomaly gradient is also overlapped by an along-axis stratification gradient produced by the Fraser River (Masson and Peña, 2009). The effect of this stratification gradient on upwelling strength is demonstrated to first order through the bulk Richardson number dependence identified in Equation 10. Since interface displacement is inversely proportional to the density difference across the interface $\Delta\rho$, we expect the weakly stratified northern SoG to experience stronger upwelling than

535 the strongly stratified southern SoG, especially near the Fraser River plume. In fact, we observe the surface nitrate anomaly in the EOF mode 3 spatial pattern to vanish nearly completely within approximately 20 km of the Fraser River mouth.

For comparison, the nitrate PCA mode 1 EOF that we attributed to upwelling along the Vancouver Island coast does not exhibit such striking along-axis asymmetry (Figure 6d). The greater uniformity of this anomaly may simply be due to the weaker overall stratification and nitracline present during the shoulder seasons. However, another consideration is that the

540 along-axis interface tilt described above is oriented in the opposite direction during the southeastly wind that drives upwelling along Vancouver Island, and if present, would now deepen the pycnocline along the downwind coastline and mitigate the effects of the weaker stratification in the north.

The second feature of the nitrate mode 3 EOF not fully explained by cross-axis upwelling is the surface nitrate anomaly throughout the western Discovery Islands (Figure 6f). While significantly weaker than the coastal upwelling anomaly, this

island signal none-the-less warrants special consideration given the ecological importance of the Discovery Islands habitat. At the southern opening of these island channels, the nitrate anomaly appears to be a northern extension of the cross-axis upwelling signal. Inside the channels however, the nitrate anomaly is more uniform and less visibly associated with any particular shoreline. A previous analysis of surface nitrate using SalishSeaCast in this region found this wind-driven nitrate signal to contribute significantly to summer monthly standard deviations during 2016 and suggested the most likely source was advection

of tidally-mixed nitrate from the northern passages (Olson et al., 2020). Given the high median surface nitrate that we observe in the northern passages of the Discovery Islands from our hindcast analysis (Figure 6a), we concur that southward advection of this persistent surface nitrate reservoir is a more accessible source of surface nitrate to the southern Discovery Islands than vertical mixing alone.

## 4.3 Comparison to other study regions

As mentioned in the introduction, the upwelling solutions discussed above in Section 4.1 have been clearly demonstrated in several well-studied systems across a range of dynamic widths. Specifically, cross-axis upwelling (Equation 10) dominates in dynamically wide basins like the Baltic Sea sub-basins (Bednorz et al., 2019) and in the North American Great Lakes (Plattner et al., 2006) while along-axis upwelling (Equation 13) dominates in dynamically narrow basins like the glacial reservoirs of British Columbia (Stevens and Lawrence, 1997). In the SoG where $L_R$ approaches the basin width, the dominant upwelling

regime is less predictable. For context, the SoG is only narrower than Lake Ontario or the Gulf of Finland by approximately half. However, the SoG is also significantly more stratified due to the high salinity gradients and thus has a significantly larger $L_R$. Ultimately, $L_R$ in the SoG is never larger than half the basin width, and upwelling falls clearly into the cross-axis regime of a large basin according to the PCA results. The $L_C/L_R = 1$ transition has been explored in fjords using an infinite channel

approximation (Cushman-Roisin et al., 1994), but this exercise neglects the importance of the corner divergences discussed earlier. Given the open passages at the ends of the SoG, perhaps an infinite channel approximation is indeed appropriate.

A striking contrast between the wind-driven pycnocline displacements in the SoG relative to the other systems discussed is the absence of basin-scale internal waves or seiches in our PCA results. Kelvin waves are an established wind response in large lakes including Lake Ontario (Csanady, 1977), Lake Geneva (Bouffard and Lemmin, 2013), Lake Iseo (Valerio et al., 2012) and Lake Tahoe (Roberts et al., 2021) while the presence of along-axis seiches in narrow lakes is well documented (Laval et al., 2008; Stevens and Lawrence, 1997). This absence of detectable wave responses is possibly due to enhanced dissipation or damping as waves travel along the irregular coastline along the entire circumference of the SoG. Coastline irregularities have been demonstrated to significantly attenuate Kelvin wave amplitude and phase speed along continental margins (Mysak and Tang, 1974) and complete damping of along-axis seiches has been documented in the Nechako Reservoir (Imam et al., 2013). More conclusively, SalishSeaCast simulations using a smooth geometric bathymetry produce Kelvin wave-like rotating seiches excited by the cross-axis upwelling response that are absent when the realistic bathymetry is used (simulations not shown).

## 4.4 Ecosystem implications

Our findings suggest that surface nitrate supply in the SoG during the productive season is primarily sourced along the eastern and western coastlines and is stronger in the north relative to the south. While phytoplankton in the SoG are extensively sampled and studied, finding consistent patterns between observed phytoplankton biomass and these surface nitrate anomalies presents several challenges. One such challenge is that seasonal chlorophyll averages are already unevenly distributed due to the along-axis stratification gradient. Specifically, surface chlorophyll is generally higher in the strongly stratified southern SoG and generally lower in the weakly stratified northern SoG (Suchy et al., 2019; Masson and Peña, 2009). This surface chlorophyll gradient alone is opposite what we would expect given the enhanced northern nitrate upwelling revealed in this study. However the chlorophyll depth distribution is also larger in the northern SoG, with the highest depth-integrated biomass between Texada Island and Vancouver Island during the summer (Masson and Peña, 2009). More recently, an analysis of 5 years of sampled nitrate and chlorophyll profiles collected as part of the Pacific Salmon Foundation (PSF) Salish Sea Marine Survival Project revealed no significant along-axis spatial trends at the seasonal level (Pawlowicz et al., 2020). Given the comprehensive spatial and temporal coverage of this data set, this finding suggests that the northern-intensified upwelling that we observe does not significantly affect chlorophyll at the seasonal level. Rather, an event-based analysis of such a data set is likely necessary to provide additional detail into the along-axis chlorophyll response to wind-driven nitrate upwelling.

A second challenge is that nitrate upwelling events are episodic by nature and strongest near the shoreline, while nutrient and chlorophyll sampling are historically cruise-based and generally clustered toward the central axis of the SoG (Masson and Peña, 2009). Such a sampling program is effective for tracking seasonal cycles and interannual variability throughout the SoG, but may entirely miss systematic chlorophyll anomalies that appear during nitrate upwelling along the coast. More spatially comprehensive data sets like the PSF data set present an opportunity to resolve these chlorophyll anomalies, however no analysis of this data set with an emphasis on the nearshore areas of the northern SoG has been completed to date. Perhaps

these upwelling-driven chlorophyll signals can be resolved in future sampling programs and modeling studies that specifically target these areas of enhanced nitrate upwelling.

Although nitrate is used as the primary upwelling tracer in this study, sub-pycnocline water is also associated with aragonite undersaturation and is thus potentially stressful for sensitive organisms such as shellfish. The two regions of particularly concentrated shellfish aquaculture activity identified in this study, Baynes Sound and the Discovery Islands, are both partially impacted by the wind-driven surface nitrate mode 1 and 3 EOF anomalies. Baynes Sound in particular is centrally located along the Vancouver Island upwelling region (Figure 6d), and we reasonably expect this area to be impacted by sub-pycnocline water

during southeasterly wind events. Conversely, the Discovery Islands are located outside of the main upwelling areas and we have stated previously that the nitrate anomalies that we observe in the EOF spatial fields in this region are likely the result of surface advection. Continuous sampling at a shore station on Quadra Island maintained by the Hakai Institute has demonstrated rapid changes in temperature, salinity and aragonite saturation state immediately following strong northerly wind events (Evans et al., 2018). While the authors determined that wind energy input was sufficient to vertically mix the water column to the depth

required to produce these changes, given the persistent southward surface velocities that accompany these wind events (Olson et al., 2020) we still argue that lateral advection likely plays a significant role in these water property changes.

## 4.5    Limitations

The dominant PCA modes that we considered for surface temperature and nitrate accounted for less than half of the total variance associated with each de-trended tracer record, with the remaining variance spread across many higher modes of no

clear physical attribution (Fig. S3). Additionally, the dominant modes that we did examine contained significant energy at the tidal frequencies regardless of attribution. Ultimately, the PCA method only captured approximately 45% of the variance into easily interpretable modes and did not separate all of the tidal variability into higher modes only. In experiments where PCA was performed on records that were lowpass-filtered to remove the tides in addition to the de-trending step, the fraction of explained variance for nitrate modes 1-3 increased to approximately 60%. This increase suggests that some of the unresolved,

higher-mode nitrate variance is indeed tidal, however a similar increase was not achieved for temperature.

        Since we have limited our analysis to surface wind, nitrate and temperature fields, we cannot draw further conclusions about the density structures that we have proposed to explain our PCA results, specifically whether the along-axis pycnocline tilt required to enhance the strength of upwelling in the north actually occurs or whether an along-axis stratification gradient indeed significantly affects the strength of upwelling. Additionally, we have neglected other geographic features that have been

suggested to influence upwelling strength such as cross-shore bottom slope (Choboter et al., 2011; Lentz and Chapman, 2004). For example, along-shore variations in bottom slope have been implicated in observations of spatially persistent upwelling hotspots in the Gulf of Finland (Delpeche-Ellmann et al., 2018). Analysis of the density structure presents a logical next step for addressing these remaining questions.

        From a more biological perspective, we have also neglected to quantify the primary productivity associated with the surface

nitrate sink and instead assumed an overly simplistic pathway between nutrient availability and phytoplankton consumption. One potential consequence of this assumption relates again to the along-axis stratification gradient and the chlorophyll depth

distribution that arises. Any complex interactions in the NPZD model involving light limitation and variability in growth and grazing rates that accompany this along-axis chlorophyll distribution and their effects on nitrate consumption would be overlooked by our study. Previous studies that identify elevated chlorophyll biomass near regions of enhanced surface nutrient supply in the SoG such as tidal jets (Olson et al., 2020) and fronts (Parsons et al., 1981) suggest that upwelling-sourced nitrate would indeed boost primary productivity as well, and we thus find our results promising. However, an integrated biophysical study that considers the fate of upwelled nitrate is another logical next step in addition to exploring the density structure.

## 5 Conclusions

We analyzed 5 years of hourly, high resolution surface nitrate and temperature results from the SalishSeaCast biophysical coupled model along with hourly, high resolution surface wind forcing fields from the operational Canadian HRDPS weather model in order to better characterize the mechanism of wind-driven surface nutrient delivery in the SoG. We found the dominant HRDPS surface wind patterns to be oriented primarily along the main axis of the SoG, with strong southeasterlies dominating in the winter and weaker northwesterlies dominating in the summer. These primarily along-axis winds produce episodic upwelling along the Vancouver Island coast during the spring and fall and along the British Columbia mainland coast during the summer. This upwelling response produces clear surface nitrate and temperature anomalies along these coasts.

Using principal component analysis (PCA), we determined that the cross-axis upwelling patterns in surface nitrate account for a combined ~30% of the variance across the 5 year record during the productive season between the spring and fall phytoplankton blooms. By sharp contrast, nearly half of the surface temperature variance over the same period is dominated by a single combined mixing and diurnal heating-cooling pattern. The power spectra of these principal component (PC) loadings time series along with the spectral coherence and correlation between the PC loadings and the spatial median of along-axis wind stress calculated from the HRDPS wind record were used to confirm these physical attributions. For surface nitrate, a positive coastal nitrate anomaly band occurs to the left of the wind stress direction. For surface temperature, strong wind stress in either direction occurs with cool anomalies while weak wind stress occurs with warm anomalies. We attribute this contrast between surface nitrate and temperature to a deeper nitracline relative to the thermocline. Under northwesterly winds, the nitrate upwelling anomaly along the eastern coastline is significantly stronger in the northern SoG relative to the south.

The cross-axis upwelling response revealed by the PCA results is consistent with basins that are wider than the baroclinic Rossby deformation radius, and contrasts with the end-to-end pycnocline setup observed in narrow lakes. However, an along-axis pycnocline tilt is theoretically possible in the SoG due to a divergence at the upwind corner of the basin. Such a tilt would be consistent with the along-axis nitrate anomaly gradient along the eastern shoreline, although the background along-axis stratification gradient driven by the Fraser River could also explain this northern intensification of the surface nitrate anomaly. Regardless of the mechanism, this enhanced upwelling has implications for summer productivity in the northern SoG. Sampling biases toward the open water regions of the SoG may prevent the impacts of this nitrate upwelling source from being observed. Overall, the episodic nature of upwelling in the SoG may disrupt the background state of coastal habitats,

whether from a nutrient perspective for bottom-up control of higher trophic levels or from a $pCO_2$ perspective for sensitive shellfish aquaculture regions such as Baynes Sound and the Discovery Islands.

*Code and data availability.* All postprocessing and analysis scripts are available from the SalishSeaCast GitHub repository (https://github.com/SalishSeaCast/SoG_upwelling_EOF_paper). SalishSeaCast and HRDPS results are available from the SalishSeaCast ERDDAP server (https://salishsea.eos.ubc.ca/erddap/griddap/index.html). All observational data used in this study are available online from their respective organizations: Environment and Climate Change Canada (ECCC) meteorological station observations (https://climate.weather.gc.ca/historical_data/search_historic_data_e.html), Fisheries and Oceans Canada (DFO) buoy observations (https://www.meds-sdmm.dfo-mpo.gc.ca/isdm-gdsi/waves-vagues/data-donnees/index-eng.asp) and NASA MODIS Aqua images (https://oceancolor.gsfc.nasa.gov/cgi/browse.pl?sen=amod). More information about SalishSeaCast can be found on the project web page (https://salishsea.eos.ubc.ca). Meteorological observing platform data may also be found via the Canadian Integrated Ocean Observing System (CIOOS, https://cioospacific.ca)

.

*Author contributions.* BMM performed the analyses and drafted the manuscript. SEA performed the hindcast simulations of SalishSeaCast. Both authors contributed equally to the development of the research concept and the manuscript beyond the initial draft. BMM is supervised by SEA.

*Competing interests.* The authors declare that no competing interests are present.

*Acknowledgements.* This work was funded by the Marine Environmental Observation, Prediction and Response (MEOPAR) Network of Canada (grant numbers 1.2, 7.2 and 37.1). Computational resources were provided by Compute Canada for hindcast runs (grant numbers FT520, RRG2648 and RRG2969) and Ocean Networks Canada for daily nowcast runs. The SalishSeaCast software environment was developed by Doug Latornell, and the SMELT ecosystem model was developed by Elise Olson. Additionally, Vy Do performed a significant amount of the preliminary analysis leading to this study as an undergraduate co-op student. We also thank Rich Pawlowicz at UBC for his assistance with the spectral analysis, and Jennifer Jackson at the Hakai Institute and one anonymous reviewer for their helpful comments on the manuscript.

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
