# Peer review of "Wind-driven upwelling and surface nutrient delivery in a semi-enclosed coastal sea"

_Ocean Science, 2021_

## Author Response (AR1)

(line numbers refer to the marked-up manuscript)

**Reponse to Anonymous Reviewer 1**

*Moore-Maley et al. (2021) use output from a biophysical model to study wind-driven upwelling in the Strait of Georgia (SoG). The authors show that the predominant wind forcing is along the axis of the Strait, and use EOF analysis to reveal cross-strait "modes" in the variability of surface nitrate concentration (and, to a lesser degree, surface temperature). They go on to relate the variability of the cross-strait modes with that of the along-axis wind forcing. In the discussion section, the authors consider the response of a two-layer model of a rectangular basin and use that to frame a discussion about cross- and along-axis symmetry as a response to wind forcing.*

*The paper is commendably well-written. The introduction section is excellent, the graphics are nice, and the discussion and conclusion sections are for the most part clearly formulated. The choice of EOF analysis as the main diagnostics tool is appropriate in principle - although I think the methodology could be improved in several ways to strengthen the paper.*

*The main weakness of the paper is pointed out by the authors themselves (Section 4.5): The analysis rests on the surface expressions of nitrate and temperature, both of which are affected by a multitude of factors and thus are complicated and indirect proxies for upwelling. It is therefore not surprising that the relationship between wind forcing and cross-axis empirical modes in fact appears relatively weak. It is my opinion that the main conclusion of the paper would be substantially strengthened by including the analysis of one or more variables more directly related to the dynamical process of upwelling. If the basic mechanism were established, the following discussion of surface nitrate and temperature would be strengthened substantially (or, if not established, that would raise some interesting questions). Conversely, if this is not done, the authors should add clear modifiers to statements such as "[this study] explicitly identif[ies] wind driven upwelling in the SoG" (line 83).*

*Nevertheless, I generally found this paper and the analysis within interesting and well-reasoned, and I believe it could constitute a significant contribution to the understanding of nutrient availability in the upper waters of the SoG, with implications for the understanding of ecosystem functioning in the area. The paper could also contribute meaningfully to the understanding of wind-driven upwelling in enclosed bodies of water more generally. I therefore recommend that this paper be published after appropriate revisions.*

We are grateful to Reviewer 1 for their thoughtful and detailed comments on this manuscript, and we have included responses below addressing each of the specific comments. As pointed out in the summary above, our initial manuscript submission based conclusions heavily on the PCA of surface nitrate and temperature without a robust method for attributing the PCA modes to physical phenomena. We appreciate the suggestion to analyze the density structure and currently have a manuscript in preparation that explores this particular research question. With respect to the present paper, we prefer to keep the study within the scope of surface nitrate variability. We have therefore focused our revisions on improving the mode attribution aspect of the PCA. Specifically, we have implemented the following changes:

- Removed T modes II and III since they account for the smallest variance. We now directly contrast T mode I vs N modes 1-3. These results are presented in the new Fig. 6.

- Improved spectral analysis of the PCs with multitaper variance reduction and coherence. We thank Reviewer 1 for their request for variance reduction and suggestion of block averaging, which inspired this new analysis. We considered using block averaging (Welch's method), but ultimately chose the multitaper method to preserve the subtidal resolution. The fortnightly tidal peaks of N mode 2 are now clearer and the subtidal wind energy is clear in the other modes relative to N mode 2. Additionally, coherence reveals the frequency separation between the higher frequency storm forcing of N mode 1 and the summer wind forcing of N mode 3, which is consistent with a Stahl et al. (2006) cluster analysis of synoptic types referenced in Section 2.1 at line 155. These results are presented in the new Fig. 8.

- Improved correlation analysis between the PCs and the wind stress, using Bayesian linear regression to determine the normal distributions and confidence intervals of the fit parameters. All presentation of correlation is now quantitative, and differences between wind-driven mixing and wind-driven upwelling are discussed in terms of the "symmetry" of each PC mode between positive and negative along-axis wind stress. These results are presented in the new Fig. 9.

- Changes in the text:

  - Clearer background on physical drivers in the SoG: fortnightly tides at line 122, synoptic forcing types and time scales beginning at line 146.
  - Presentation of spectral analysis and correlation methods beginning at line 271
  - Definition of mode attribution criteria beginning at line 293
  - Results presented in terms of mode attribution criteria throughout Section 3.2
  - Mode attributions summarized beginning at line 507

Additionally, we have clarified our statement in the final paragraph of the introduction about the contribution of this study at line 93, and we have expanded on the limitations of the study beginning in Section 4.5 at line 760 to more clearly state the next steps. We are confident that these revisions along with the line-by-line changes below will satisfy the concerns raised by Reviewer 1.

**Major comments**

1. *The authors base their analysis on the surface expressions of temperature and nitrate. For both parameters, it is hard to separate the effects of upwelling, lateral advection, and wind-driven mixing, and both parameters are presumably strongly influenced by tides and diurnally varying surface forcing. While I understand that the authors wish to limit the scope of their study, I think the main conclusion (wind-driven upwelling plays a major part in the dynamical response to wind forcing events and their effects on nutrient distribution) would be much stronger if the authors included an analysis of upwelling more directly. An advanced methodology would not necessarily be required, and relevant parameters should be available to the authors in the model output. For example, pycnocline/isopycnal depth at select locations along the respective sides could be used in a simple comparison with along-axis winds, or they could use cross-axis isopycnal tilt at an appropriate cross-section. The point would be to more conclusively establish the asymmetric upwelling pattern as a response to wind forcing events before going on to the more detailed spatial analysis and more complex discussion of mechanisms. It would also in my view tie the theoretical framework of section 4.1 more together with the results of the study.*

   As we mentioned above, we have chosen to keep the surface nitrate focus of the study and improve on the mode attribution aspect of the analysis. To that end, we have improved the spectral analysis and correlation methods used and reframed the presentation of the PCA results in terms of physical mode attribution based on these new methods. Using this new approach, we clearly demonstrate that nitrate modes 1 and 3 are upwelling modes and together account for 30% of the subtidal variance in the interior SoG. Specifically, these modes are sufficiently asymmetric in their correlation to directional along-axis wind stress to rule out wind mixing, and the improved power spectra clearly show the energy shift toward subtidal frequencies relative to Temperature mode I. The new methods are described in Section 2.3 starting at line 271, and the revised results are shown throughout Section 3.2 starting at line 415 and in Figs. 6-9.

2. *Figures 5/8 and the discussion of spectra: The spectra as they are are very noisy, and it is currently difficult to discern peaks that are central in the description and analysis. I strongly suggest using block averaged spectra (the record lengths should be more than sufficient to do so). This should make the signals of interest clearer with respect to the background noise while still resolving the entire frequency band of interest. I also suggest including error bars on the spectra given that minor peaks are given significance in the interpretation.*

   We have redone the spectral analysis using the multitaper method to reduce the noise in the power spectra and described the method beginnning at line 271. We have also removed all spectral analysis from the raw data and focused instead on just the PC loadings. This revised analysis clarifies the fortnightly peaks in nitrate mode 2 and shows the broader subtidal energy in the remaining modes relative to nitrate mode 2. A subtidal peak in nitrate mode 3 centered at the 15-20 band is also revealed (Fig. 7), which is consistent with a Stahl et al. (2006) cluster analysis of synoptic types referenced in Section 2.1 at line 155.

3. *In general, the relationship between EOF indices and winds do not strike me as obviously strong, neither in figure 7 nor in the correlation analysis. This does not mean that the wind-driven upwelling mechanism suggested by the authors does not occur - especially given that tides and diurnal*

*variability in surface forcing likely adds a lot of "noise" and may affect the mode structures in different ways. I wonder if the authors would be better off focusing explicitly on the \*subtidal\* variability in their EOF analysis - rather than performing the EOF analysis on full-resolution data and then "detiding" indices and winds when looking at correlations. It seems to me that the dynamically relevant time scales are all longer than a day: wind pulses and ocean responses both appear to have longer time scales (hence the filtering in figures 4 and 7?), and the response time of the upwelling process is also shown by the authors to be more than one day. The authors already do the same at the opposite end of the spectrum by applying a 50 d high-pass filter. Could appropriate filtering to explicitly focus on subtidal signals restrict the analysis to the frequency band of interest - and make the results of the EOF analysis easier to interpret?*

We have modified our qualitative presentation of the raw hindcast time series (Fig. 5) and the PC time series (Fig. 7) by removing the non-upwelling signals and zooming the time axis. These figures now show the overlap between wind events and surface anomalies more clearly. More importantly, we have improved the quantitative element of the PC wind dependence by including the significance of the spectral coherence (Fig, 8) and linear regression fits (Fig. 9) between the PCs and the along-axis wind stress, based on 99% confidence intervals. Nitrate modes 1 and 3 are significantly coherent and correlated with along-axis wind stress in directions and frequency ranges that support our upwelling mode attribution, and we are confident that the revisions show this result clearly.

4. *Line 312-314: I have some difficulty seeing this described relationship between the mode loading time series and the winds in Fig 7. I suggest giving the reader some more specific pointers in the text, and perhaps indicating "spikes" in Fig 7.*

We have zoomed the time axis and removed all modes except N modes 1 and 3. The overlap between the PCs and the wind is now clear (Fig. 7). Fig. 7 is also now directly comparable to the raw time series in Fig. 5 since we have removed the tidal mixing signals and zoomed the time axis. We mention this comparison in the text at line 442.

5. *Some sort of characterization of the time scale of wind events is needed - perhaps around Line 273. A typical time scale is alluded to later in the manuscript (Line 395, 404), but never really described based on observations or literature. I would suggest adding a description of the typical duration of both wind events (from HRDPS) and upwelling events (from the ocean model).*

We have added a synoptic summary of the NE Pacific based on literature, including time scales, in Section 2.1 beginning at line 146. These time scales are consistent with the coherent bands that emerge between the HRDPS along-axis wind stress and nitrate PC modes 1 and 3 in Fig. 8. Additionally, the lag time between a wind event and an upwelling response is now better characterized according to the peak-to-peak lag in Fig. 8 in addition to the back-averaging windows in Fig. 9.

6. *Section 4.1: While I found the theoretical exploration of the two-layer basin useful, I wonder if the SoG doesn't deviate from the model in another fundamental way: It seems to me that since the the SoG is \*not\*, in fact, closed at either end, along-axis pressure gradients may not be able to build up to the degree implied by the example. I think this warrants some discussion, most likely in Section 4.2.*

We added a paragraph beginning at line 644 that summarizes this statement.

**Minor comments**

1. *Line 3: It should be made clear that the skilled reproduction of observations of all these parameters are not shown in this study, but come from previous work.*

Clarified at line 4.

2. *Line 7: "climatology" is a little confusing - suggest rephrasing to "predominant wind pattern" or simply "Alongaxis winds steered..".*

Changed at line 7 and throughout.

3. *Line 30: "Basin scale" here should be replaced by "dynamical width" or similar.*

Changed at line 31 and throughout.

4. *Line 92-94: Please provide a reference for the estuarine circulation/exchange.*

Reference to MacCready et al. 2021, JGR at line 104.

5. *139: "partial steps at the bottom boundary" - the meaning of this is not clear to me. Please clarify.*

This statement is unnecessary modelling detail. Removed at line 170.

6. *Line 146: In Section 2.2, the model is described, including the configuration of the biogeochemical parameters (silica, plankton species etc). Most of these are never mentioned again. If they have little influence on the results, the authors may want to mention that here. If not, the authors should at least mention in the discussion section how the biogeochemical components of the model might play into the results presented here (does biological consumption end nitrate spikes, for example?).*

We have added a sentence beginning at line 184 that states the importance of all NPZD functional groups for resolving the post-bloom surface nitrate sink.

7. *Line 181: Please indicate (roughly) the timing of the freshet.*

Defined at line 118.

8. *Line 260: "Provides significant physical driver" - this statement should be qualified to be less strong (perhaps include a "potentially" or similar?). Also rephrase (" *a* significant driver?").*

Clarified at line 355.

9. *Line 263: Please explain why these particular locations were chosen. For example - why the Texada spot and not a spot across from Central VI?*

We broadened these location definitions to be spatial medians over the entire upwelling coastlines. The coastal sections are shown in the new Fig. 4 and described in the paragraph beginning at line 358.

10. *Line 272: "Averaged over the SoG region": Please be a little more specific about what area winds were averaged over.*

We defined an open water region in the SoG shown in the new Fig. 4. All spatial medians of surface wind, nitrate and temperature have been revised to refer back to this region and the coastal transects also shown in Fig. 4. For example, the open water medians of nitrate and temperature are first mentioned at line 362 and the open water median of HRDPS wind speed is first mentioned at line 376.

11. *Line 297: I find the use of "low-frequency" here unclear - especially since it seems to include the diurnal band. Please clarify.*

We removed the spectral analysis from Section 3.1, so this statement has been removed as well. However, our later presentation of the spectral analysis of the PC modes has been revised to use "subtidal" rather than "low-frequency" where appropriate, for example at lines 457, 470.

12. *Line 303: "which represent" should be qualified (e.g. "which we interpret to represent").*

We have removed this statement and attribute physical processes to each mode later beginning at line 507, once our mode attribution criteria have been satisfied.

13. *Line 306/328: Temperature mode III also seems to have a strong N-S structure, but it is consistently referred to as a cross-axis mode. Please clarify/comment.*

We have removed Temperature modes II and III from the analysis since the variance fraction they each explain is less than 10%.

14. *Line 330: "time-averaged" here is confusing - reads as an average across all time points. Please rephrase.*

We have rephrased this statement as a "back-average" at line 489, and added a definition in Section 2.3 at line 291.

15. *Line 334: "small amount of correlation" : confusing, please rephrase.*

We have removed this statement and now discuss the correlation strength in terms of the regression slope and $R^2$ separately for positive and negative along-axis wind stress. These revised results are summarized in the new Fig. 9 and in the paragraphs beginning at lines 488 and 498.

16. *Line 341: "Visibly correlate" - I suggest avoiding this terminology if no significant correlation was found in the quantitative analysis.*

We have removed this statement. All references to correlation now use the linear regression fit parameters shown in the revised Fig. 9.

17. *Line 367: Please explain briefly which assumptions have gone into transforming Ri(U) to Ri(tau).*

We write Ri(tau) assuming that the shear stress between layers translates directly from the surface wind stress. This assumption is common in two-layer models of seiching in lakes. Additionally, we use Ri as an upwelling scale parameter only and not an indicator of turbulence (which we have neglected here). We have added a clarifying statement at line 567.

18. *Line 374 & 376: Surely "the coasts" are always important? Please rephrase.*

We have removed this statement. Our revised framing of this discussion contrasts "infinite channel" solutions with a closed box. The effect of the closed box is manifested in the "corner divergences" introduced beginning at line 620. We have added a new figure (Fig. 10) to demonstrate this concept. We also qualify this model with the existence of channels at the ends of the basin in the paragraph beginning at line 644.

19. *394-396: There is an apparent contradiction here - should the conclusion not be the opposite? Please clarify*

This was a typo, fixed at line 604.

20. *Line 414-415: "If the cross-axis..fluxes": This is not self-evident to me. Please add some explanation or a reference.*

We have clarified this argument as a direct interaction between the cross-axis pressure gradients that results from the corner divergences and the cross-axis Ekman fluxes. This interaction is summarized by the blue and black arrows in Fig. 10c. The blue pressure gradient arrows are not explicitly resolved by the 2-layer solutions, we instead infer them from the corner divergences and reason that they will change the surface force balance to allow downwind surface transport in the interior of the box.

21. *Figure 1: Please add a scale bar. I would also suggest changing the color of the Texada star marker as it currently disappears into the background a bit.*

We have added a scalebar and removed the Texada marker, which is now replaced with a transect of grid points along the eastern shore in Fig. 4.

22. *Figure 4: It is difficult to see the wind time series here. I also find that much of what is in the figure caption belongs in the text proper.*

We have removed the tidal mixing signals and zoomed the time axis to improved the reader's ability to interpret this figure. We have also removed the last several sentences of the caption. This figure is now Fig. 5.

23. *Figure 4: Please indicate the timing of the snapshots shown in Figure 2.*

We have added green lines to Fig. 5 to show these times. We have also added a statement at line 385 comparing this figure with the snapshot Fig. 2.

24. *Figure 5: Spectra should probably be computed based on the productive seasons only - as for the profiles above.*

We have removed all spectral analysis of the raw SalishSeaCast records in Section 3.1.

**Technical corrections**

1. *Figure 6 needs a length scale. Could be achieved by using length instead of grid coordinates on the x/y axes (grid coordinates are not very useful in any case).*

We have added a scalebar to Fig. 6.

2. *Line 239 - 240: I recommend using a standard date format - the Ocean Science convention seems to be "25 July 2007".*

We have corrected the date format in Fig. 2 and the caption, as well as at line 333.

3. *Line 551: "right" −> left?*

   This was a typo. Fixed at line 810.

4. *Figure 2: It would be useful to include an indication of the predominant wind direction above each 3 plots. Maybe using some simple arrows arrows or adding direction in text at the top axis title.*

   We've added the wind direction as text in the titles of Fig. 2.

5. *Figure 5: Could the difference between the colors used for the median profile and IQ range be made a little stronger in a and d? Currently a little difficult to see the median profiles.*

   We now only show profiles for spatial medians of nitrate and temperature over the open water region in Fig. 4a. The colors are now red for nitrate and black for temperature in Fig. 4b, which improves the contrast.

6. *Figures 5, 8: Please add units to PSD y-scales.*

   All of the PSD plots in the paper are now in Fig. 8a. We've included units of $1/\mathrm{d}^{-1}$ since the spectra are normalized by power.

7. *Figures 7, right: The sharp red color makes it difficult to discern the other time series. Please reconsider the color and/or opacity of these lines.*

   We've removed all PC modes except nitrate modes 1 and 3, which have a decent contrast between warm orange and cool blue-green.

8. *Figures 8 (bottom): I suggest changing color of the horizontal line in case of difficulties for colorblind readers.*

   We've removed the horizontal line for the 2d histograms, since it wasn't necessary.

9. *Title and elsewhere: Should "wind driven" be hyphenated ("wind-driven") since it is a compound adjective?*

   Fixed in the title and throughout.

**Reponse to reviewer 2, Jennifer Jackson**

*The manuscript by Moore-Maley and Allen uses model output from a high-resolution biophysical model to examine upwelling in the Strait of Georgia. This is an important research question. Upwelling is often discussed in the Strait of Georgia but has never been examined in detail or quantified. The authors use five years of model output (focusing on temperature and nitrate) and high-resolution wind climatology to study upwelling. In general, the manuscript is well-written and interesting and will add important knowledge about physical processes in the Strait of Georgia. That being said, I did struggle with sections 3.2 and 4 and think that considerable improvement is needed, particularly in these sections, before the manuscript is published in Ocean Sciences. I therefore recommend major revisions. Details are listed below.*

We are grateful to Dr. Jackson for her thoughtful and detailed comments on this manuscript, and we have included responses below addressing each of the specific comments. As mentioned above and elaborated on below, our initial manuscript did not sufficiently describe the process for PCA mode attribution to specific physical phenomena, and several important steps were overlooked. We have made significant revisions to resolve these issues that are summarized in our main response to reviewer 1 above. We are confident that these revisions along with the changes listed below will satisfy any outstanding issues raised by Dr. Jackson.

**Major comments**

1. *My first major concern is with the interpretation of the principal component analysis in section 3.2. Lines 303 to 310 describe the dominant modes from the EOF spatial patterns and based on EOF results shown in Figure 6. Despite the importance of these results for the manuscript, I think that important information is missing from the description on the EOF results. This includes:*

   - *The percentage of variance calculated within each mode*

- *A description of how the modes were diagnosed (e.g. beyond a picture, how is it known that mode 1 of for nitrate is upwelling along the western shore?). Some work was done to diagnose the different modes in Figures 7 and 8 but these results were not always conclusive*

- *A description of what a mixing-heating pattern is (lines 305 to 306) and how this in particular was diagnosed.*

- *Throughout section 3.2 (and in the figures), there are several references to positive and negative variance. I don't know what positive and negative variance means in regards to these results. Please clarify.*

Responding to this issue raised by Dr. Jackson was central to our revisions, and we have summarized the relevant changes in our general response to Reviewer 1. Briefly, we have expanded Section 2.3 to describe our spectral analysis and correlation methods (line 271) and define our mode attribution criteria (line 293). We have modified Figs. 6-9 to account for these updated methods and criteria. We have revised Section 3.2 to present the PCA results in the context of this mode attribution (line 430). We then summarize our mode attribution at the end of 3.2 (line 507).

2. *My second major concern was the lack of discussion of stochastic events (i.e. storms) in the manuscript. HRDPS shown in Figure 4 shows the stochastic nature of the events that cause upwelling and downwelling, and the impact some of these events have on surface temperature and nitrate. Despite the frequency and strength of these events, there doesn't appear to be a stochastic (1 to 3 day) frequency in the power spectra on either Figures 5 or 8. If the authors are arguing that storm driven upwelling or downwelling are the dominant modes for temperature and nitrate variability in the Strait of Georgia then why don't stochastic events evident in the power spectra?*

Our updated spectral analysis methods have improved the resolution of some of these features (Fig. 8). The nitrate mode correlated with northwesterly wind (mode 3) now has a distinct peak in power spectral density in the 15-20 d band (green dashed line), and the coherence between the PC and the wind stress is also strongest in that band. The coherence between nitrate PC mode 1 and wind stress is restricted to higher frequencies, which is more consistent with winter storms. These time scales are consistent with the synoptic types presented by Stahl et al. (2006) *Int. J. Climatol.* (referenced in Section 2.1 at line 155). Nitrate PC mode 1 lacks both the pronounced subtidal peak and the strength of coherence that we observe in nitrate PC mode 3. This weaker signal is likely due to the weaker vertical gradients during the shoulder seasons when mode 1 is most active, and the fact that many of the shoulder season upwelling events are cut off by the time bounds of the productive season window. Ultimately, we consider the significant coherence of nitrate PC mode 1 in the 2.5-9 d band (Fig. 8b, blue-green line) to be the clearest indicator of the role of stochastic storm winds, but we don't draw any conclusions about the smaller mode 1 peaks in PSD or coherence within this band.

3. *My third major concern was the confusion of reading a manuscript where many mathematical symbols are used throughout. To make this manuscript easier to read, I suggest adding a table that details all of the mathematical symbols.*

The majority of symbols used in the manuscript are presented in the Methods Sections 2.2 and 2.3 and in the Discussion Section 4.1. Since these symbols are referenced only in the sections where they are presented and not in the Results Section 3 or elsewhere in the manuscript, we feel that a table may add unintended emphasis to the importance of these symbols and distract the reader. We have made 2 significant changes instead that make keeping track of the symbols more manageable:

- We have generalized the Rossby radius definition in the introduction (Equation 1) as $L_R = NH/f$, which is more appropriate than the previous 2-layer definition at this stage anyway. This change removes the connection between Equation 1 and Section 4.1 so the reader no longer has to jump back and forth.

- We have added a summary figure of the solutions presented in Section 4.1 (Fig. 10) that displays many of the symbols used.

4. *My fourth major concern is section 4.1. This was a complex section and I'm not clear exactly how it strengthened the manuscript. Specifically, I think that some discussion is needed to explain why using a 2 layer model is realistic in a such a complex region where 3 to 4 layers (e.g. Stevens et al.,*

*2021, Johannessen et al., 2014) are often observed. I suggest rewriting this section to emphasize to the reader why these case studies are needed and how they influence the results of the model. I also suggest that, if the case studies are used, the authors include figures of the results so that the case studies are easier to interpret.*

Section 4.1 provides important physical context to the coastal anomalies that we observe in nitrate EOF modes 1 and 3. To improve the purpose and clarity of this section, we have made the following changes.

- We have added a paragraph beginning at line 540 that defends our use of a 2-layer approximation for context purposes.
- We have revised the presentation of the solutions in terms of infinite channels and clarified our reasoning behind the closed box case.
- We have added Fig. 10 to guide the reader through the 3 different cases.
- We have added a summary paragraph beginning at line 644 that compares the infinite channel and closed box to the real SoG and the PCA results

5. *I think some key references are missing. These include:*

   - *Johannessen, S.C., Macdonald, R. W., and Strivens, J.E. 2021. Has primary production declined in the Salish Sea? Canadian Journal of Fisheries and Aquatic Sciences*
   - *Johannessen, S.C., Masson, D. and Macdonald, R.W. 2014. Oxygen in the deep Strait of Georgia, 1951-2009: The roles of mixing, deep-water renewal, and remineralization of organic carbon. Limnology and Oceanography 59(1): 211-222*
   - *Del Bel Belluz, J., Peña, M.A., Jackson, J.M.et al. Phytoplankton Composition and Environmental Drivers in the Northern Strait of Georgia (Salish Sea), British Columbia, Canada. Estuaries and Coasts 44, 1419–1439 (2021). https://doi.org/10.1007/s12237-020-00858-2*

   We have referenced Johannessen et al. (2021) and line 71, Del Bel Belluz et al. (2021) at lines 77, 78, 82, 120 and Johannessen et al. (2014) at lines 68, 113.

**Minor comments**

1. *Line 28 – I suggest adding references to previous research on upwelling in enclosed basins*

   We have included references throughout the second paragraph of the introduction, beginning at line 29. In response to this comment, we have added a few more general references at lines 30-31.

2. *Lines 45 to 57 – I found this paragraph confusing and it was difficult to understand the point of the paragraph. I suggest rewriting this paragraph so the point is more clear.*

   This paragraph was not central to framing the research context, so we removed it.

3. *Lines 151 to 152 – Please add a reference here*

   We have referenced Hansen et al., 2013, *Harmful Algae* at line 183.

4. *Line 165 – How realistic are these 2.5 km winds in some of the narrow channels within the Salish Sea? Do these coarse winds (relative to the complexity of the study area) impact the results?*

   Our primary HRDPS evaluation for this study is presented in the paragraph beginning at line 345. We intentionally chose open water locations because we know that HRDPS performs well there, and wind along the main channel is the primary source of upwelling. HRDPS 2.5 does not perform well in narrow inlets, for instance Howe Sound, but we do not expect wind in the inlets to have a significant impact on upwelling.

5. *Line 208 – Figure 1 includes Juan de Fuca Strait yet this states that only the region to the tidal mixing area (Haro Strait?) is considered. Please clarify.*

   We have added a red boxed region to Fig. 1 that shows the subregion used for PCA. We reference this highlighted region at lines 247 and 416.

6. *Lines 209 to 213 – As a reader it was difficult to interpret what the authors are stating here. If possible, I suggest adding this information to a figure. Otherwise, please make this information clearer so that it is easier to interpret.*

   We have revised this paragraph to improve the clarity. The revised paragraph begins at line 244.

7. *Lines 214 to 226 – Are the references at the end of this paragraph for the whole PC and EOF equations? Please clarify*

   We have clarified these references at lines 262, 269 and 270.

8. *Lines 257 to 258 – I don't understand the sentence starting with "There is also a tendency..." Please clarify.*

   We have clarified this statement at line 352.

9. *Figure 2 – The letters in the figure to identify the panels (i.e. a to d) does not match the description in the caption.*

   We have fixed the panel letters in Fig. 2.

10. *Lines 274 to 275 – I can't see this result in the figures.*

    We have zoomed the time axis and removed the unnecessary tidal mixing region curves in Fig. 5., and revised the text beginning at line 379.

11. *Lines 291 to 292 – It is really difficult to see the correlation between winds and temperature/nitrate at individual locations in Figure 4.*

    The zoomed time axis and removal of tidal mixing locations in Fig. 5 makes it easier to see how the coastal signals overlap with the wind. The text has also been revised to better describe Fig. 5, beginning at line 379.

12. *Lines 306 to 310 – As mentioned above, it is not clear to me how these interpretations were made.*

    We have completely revised this Section 3.2 to follow the mode attribution criteria presented in Section 2.3 beginning at line 293.

13. *Figure 7 – What do positive and negative winds and PC amplitude mean?*

    We have revised the caption text of Fig. 7 to remove these terms.

14. *Line 336 – How does the averaging window of 54 hours impact the storm data? In other words, does this averaging window minimize storm energy?*

    The PC loading is not really a function of instantaneous wind as much as the time-integrated wind. This dependence is demonstrated in the infinite channel solution along the sides of the basin in 4.1 since $\zeta_{side}$ is proportional to $\tau t$. The averaging window is analogous to time-integration since the average is just the finite integral divided by the window length. In this sense, the averaging process should not effect the energy imparted from a storm. We have added a sentence at line 290 that distinguishes between instantaneous and cumulative wind stress.

15. *Figure 8 – Again, what do positive and negative PC amplitudes mean? Also, Figure 8b shows significant energy at fortnightly and monthly frequencies. This EOF was interpreted as being dominated by tidal mixing. Please explain why tidal mixing would have significant energy here at these frequencies?*

    The original figure is now separated into a spectral analysis Fig. 8 and a linear regression Fig. 9. We have clarified that positive wind stress is southeasterly and negative wind stress is northwesterly in Fig. 9. The fortnightly and monthly peaks in PC mode 2 in Fig. 8 are the subtidal modulation of tidal mixing strength due to the tropical fortnightly tidal cycle. We have modified our language throughout the manuscript to emphasize that the tidal variability we are interested in is fortnightly modulation.

16. *Lines 571 – I think that much of the observational data used are available on CIOOS. I suggest that the authors add the CIOOS data link to the acknowledgements (https://cioospacific.ca).*

    We have added a link to CIOOS in the Data Availability section.

---

## Author Response (AR2)

**Response to Anonymous Reviewer 1**

*In my opinion, this version of the manuscript constitutes a significant improvement upon the initially submitted version. In the revised version of the manuscript, the authors have refined their methods and improved their description of study and its results, and the presentation of the material has become clearer as a result. The new sections at L293 and S3.2 are very clarifying in terms of how they conduct the study and interpret their results – it is now clearly laid out how the authors arrive at physical interpretations of the EOF modes, and uncertainties/limitations in these interpretations are thoroughly laid out. Throughout, the writing is clear and concise, and the findings are evaluated sceptically. The updated graphics are nice, and the added supporting figures for the analytical model are useful.*

*I maintain that for studying this kind of process, the ideal order would be to first study the "dynamical" variables (density) and then look at resulting effects on tracers - but I understand the authors' decision and accept their approach.*

*My initial concerns have largely been addressed. I would like to congratulate the authors on an excellent manuscript, which I recommend be published. A small number of minor suggestions are attached below.*

We are grateful to Reviewer 1 for their thoughtful follow-up comments on this manuscript, and we are pleased to hear that our last round of revisions satisfied the reviewer's initial concerns. We have addressed the remaining outstanding issues that Reviewer 1 has identified and included our responses below.

**Minor comments**

1. *Section 3.2: Recommend reiterating here that PCA analysis is only done for productive seasons.*

   We have added a statement clarifying that the productive season is used in the first sentence of Section 3.2 at line 351.

2. *Fig 8: Please indicate confidence intervals on spectra. Related: please clarify what is meant by "significant" at L461.*

   We would like to acknowledge this comment as it forced us to think a bit more about how we interpret the PC spectra. We concluded that meteorologically-forced signals in the context of our study follow stochastic, autoregressive "red noise" spectra due to persistence in atmospheric patterns. We then define significance as a peak which deviates from this stochastic process beyond some confidence interval. Solar and tidally-driven processes will have significant peaks at certain frequencies, but the absence of significance is actually evidence for wind-driven upwelling and mixing (although the 17d peak in nitrate mode 3 is significant). We determined the confidence interval for red noise by fitting an AR1 process to the 4 PC modes presented in the main text, and then finding the 99th percentile of 1000 random simulations. We have added this curve to Fig. 8 and have made text modifications throughout the manuscript at lines 231-235, 275-277, 376-379 and 424-431 to communicate this concept. We also introduced a supplement with companion figures to Fig. 6 and 8a (S2, S3) that include the first five modes for each tracer.

3. *Fig 8: For easy reference to the text at page 21, it would be useful to add a second x-axis with cycle periods (days), or at least label the lines in figure 8a with periods.*

   We labelled the lines in Fig. 8a with periods to avoid over-cluttering the figure.

4. *Fig 10: Recommend clarifying "infinite channel" – it took me some time to figure out the meaning of the title in 10b. Maybe specify "infinite in the x-direction" or similar?*

   We added sentences at lines 472-473 to introduce the infinite channels leading up to the next section at line 476. These sentences clarify that the infinite channels have zero gradients parallel to the channel walls.

5. *L108/148: Spell out or define BC for readers not familiar with this area.*

   We removed "BC" at line 112 and added "British Columbia" at line 167.

6. *L2/L19: hyphen?*

   We hyphenated "wind-driven" at lines 2, 19 and throughout the manuscript.

7. *L667: Please refer to figure 6f.*

   We have referenced Fig. 6f at line 537.

8. *Figure 4: Recommend making sure VI/SC points are discernible in B/W (different symbols?).*

   We changed the Sunshine Coast symbols to stars in Fig. 4.

**Response to Reviewer 2, Jennifer Jackson**

*The authors have done a meticulous job of addressing my concerns from the first round of revisions. The manuscript is now much clearer and easier to follow. I recommend the manuscript for publication. Below are a few minor suggestions.*

We are grateful to Dr. Jackson for her thoughtful follow-up comments on this manuscript, and we are pleased to hear that our last round of revisions satisfied her initial concerns. We have addressed the remaining outstanding issues that Dr. Jackson has identified and included our responses below.

**Minor comments**

1. *Evans et al., 2018 is cited but is not in the references*

   We have made sure that Evans et al., 2018 and all other citations are now included in the references.

2. *Lines 245 to 247 – I think that references are needed here*

   We modified this text to present our basis for mode attribution in steps so that we could link back to the study area background presented in Section 2.1 and then eliminate the river and biological processes before presenting our final mode attribution candidates. We added references to Fleming and Clark 2005, Pawlowicz et al., 2019 and Halverson and Pawlowicz 2016 to support our disregard of the rivers and Del Bel Belluz et al, 2021 to support our disregard of the biology. The new text is at lines 249-263.

3. *Lines 249 to 251 – I disagree with this statement – while try for rain or snow-fed rivers, glacial-fed rivers such as the Homalco and Fraser often peak in the summer.*

   We modified this statement at lines 252-254 to acknowledge that glacial rivers are indeed at high flow stage during the summer and to clarify that we do not consider the rivers as drivers of surface tracer variability because the time scales of this variability are longer than the synoptic time scales of interest.

4. *Figure 5 – Please clarify that the output shown in b to k are the average of the gridpoints in Figure 4a? Also, what is the variability of this average? It is unlikely that you can put a measure of variability on these already complicated figures but if you are calculating an average over all of these gridpoints then I think it is important to somehow show variability.*

   The lines in Fig. 5 are the spatial medians across the magenta region and blue/gold gridpoints shown in Fig. 4a. We have now clarified this detail in the Fig. 5 caption. We added a companion figure to Fig. 5 in the supplement that includes the interquartile range (IQR) as an envelope around each time series (Fig. S1) and added a sentence at lines 348-349 that links the IQR to the along-shore variability that we observe in the surface tracer images in Fig. 2.

5. *Line 347 – I am still unclear how Figure 6c is indicative of wind mixing and diurnal heating and cooling.*

   We modified the text at lines 364-366 clarifying that the spatial uniformity of the temperature anomaly in the interior SoG is our basis for suspecting wind mixing or diurnal heating/cooling, based on mode attribution criteria (1).

6. *Lines 347 to 348 – The cross-axis temperature gradient is very small and I don't see how upwelling can be inferred from Figure 6c.*

   We have removed our mention of the cross-axis temperature gradient at line 367 since it is not central to our results.